# Towards Efficient and Expressive Offline RL via Flow-Anchored Noise-conditioned Q-Learning

Sungyoung Lee [1]   Dohyeong Kim [2]   Eshan Balachandar [1]   Zelal Su Mustafaoglu [1]   Keshav Pingali [1]

## Abstract

We propose Flow-Anchored Noise-conditioned Q-Learning (FAN), a highly efficient and high-performing offline reinforcement learning (RL) algorithm. Recent work has shown that expressive flow policies and distributional critics improve offline RL performance, but at a high computational cost. Specifically, flow policies require iterative sampling to produce a single action, and distributional critics require computation over multiple samples (e.g., quantiles) to estimate value. To address these inefficiencies while maintaining high performance, we introduce FAN. Our method employs a behavior regularization technique that uses a single flow policy iteration and requires a single Gaussian noise sample for distributional critics. Our theoretical analysis of convergence and performance bounds demonstrates that these simplifications not only improve efficiency but also lead to superior task performance. Experiments on robotic manipulation and locomotion tasks demonstrate that FAN achieves state-of-the-art performance while significantly reducing both training and inference runtimes. We release our code at https://github.com/brianlsy98/FAN.

## 1. Introduction

Offline Reinforcement Learning (RL) (Lange et al., 2012; Levine et al., 2020) aims to learn a policy using only a fixed dataset of pre-collected interactions. This allows for the safe and efficient reuse of large historical data; however, the lack of online feedback prevents the agent from correcting errors, making it prone to value overestimation for actions outside the dataset state-action (behavior) distribution (Fujimoto

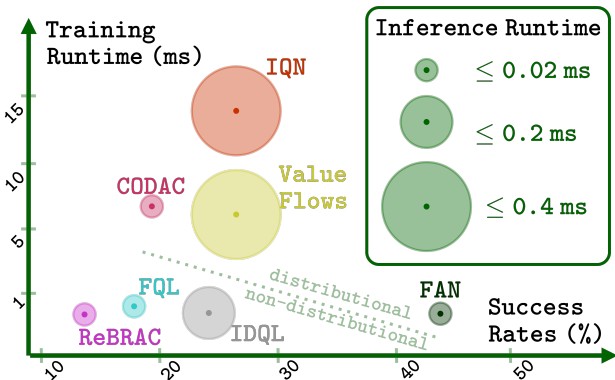

*Figure 1.* **Training Runtime per Batch vs. Average Success Rates** on five OGBench `puzzle-4x4-singleplay-v0` tasks. FAN performs the best with the highest computational efficiency.

et al., 2019; Kumar et al., 2019). Therefore, a core challenge in offline RL lies in maximizing returns while constraining the learned policy to the behavior policy that generated the data. For effective constraints, recent work has adopted expressive algorithms for learning the policy and the value.

First, flow matching has been widely used for policy training (Espinosa-Dice et al., 2025b; Wang et al., 2026). Unlike Gaussian-based approaches that are limited to unimodal distributions, flow policies can learn complex and multimodal dataset behaviors (Park et al., 2025c; Nguimatsia Tiofack et al., 2026). This enables more expressive constraints for the policy, allowing it to outperform the offline dataset behavior (Espinosa-Dice et al., 2025a; Park et al., 2026).

Second, there have been approaches using distributional critics (Ma et al., 2021; Dong et al., 2026), which learn the distribution of returns. These critics capture information that cannot be fully represented by expected returns, *e.g.*, return uncertainty. This distributional expressivity is often achieved by modeling multiple statistics of the distribution via quantiles (Dabney et al., 2018a;b), which represent the cumulative probability thresholds of the return distribution.

In this work, we focus on the *computational efficiency* of these expressive training mechanisms. As seen in Figure 1, methods employing behavior flow policies and distributional critics are computationally expensive. First, flow policies require multiple forward iterations to produce a single action, which increases the computational cost proportionally

[1]The University of Texas at Austin, Austin, TX, USA [2]Independent Researcher, Seoul, South Korea. Correspondence to: Sungyoung Lee <sylee@utexas.edu>, Keshav K. Pingali <pingali@cs.utexas.edu>.

*Proceedings of the $43^{rd}$ International Conference on Machine Learning*, Seoul, South Korea. PMLR 306, 2026. Copyright 2026 by the author(s).

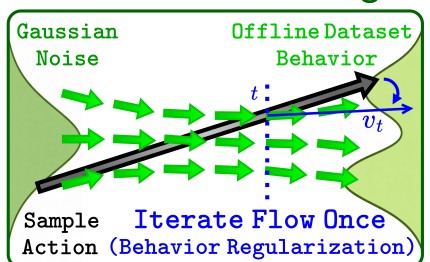
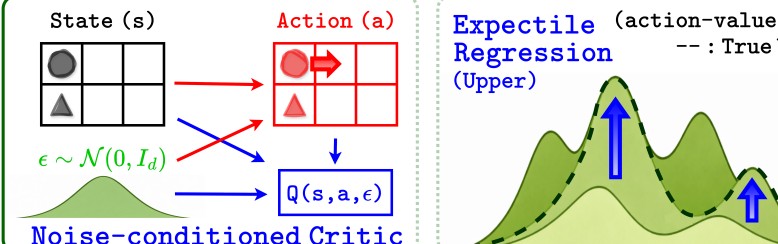

*Figure 2.* **Overview of FAN.** (*Left*) Behavior regularization utilizes only a single flow policy iteration and is applied to both actor and critic updates. (*Middle*) The distributional critic is conditioned on the same noise used for policy sampling. (*Right*) The critic update incorporates an upper expectile regression to capture maximum possible distributional returns.

to the number of flow steps. Second, distributional critics necessitate processing multiple samples (*e.g.*, quantiles), scaling the cost linearly with the number of samples. This motivates the main question explored in our study:

*How can we leverage flow policies and distributional critics to achieve state-of-the-art offline RL performance, while simultaneously improving computational efficiency?*

Specifically, we investigate whether (1) behavior flow policies can remain effective with a single flow iteration, and (2) distributional critics can be trained using a single sample.

To this end, we propose **Flow-Anchored Noise-conditioned Q-Learning (FAN)**. First, FAN utilizes flow policies but restricts them to a single iteration for behavior regularization, a mechanism we term *Flow Anchoring*. Second, FAN employs a *Noise-conditioned Critic*, which captures distributional return information while being trainable using a single Gaussian noise sample. This critic is defined by the proposed operator $\mathcal{T}_n^\pi$ in Eq.(9), which tightly couples the policy and value functions through shared noise inputs. Experiments on D4RL and OGBench demonstrate that FAN achieves best or near-best performance while reducing training runtime by at least $5\times$ compared to prior distributional approaches. Furthermore, its inference speed is among the fastest, competitive even with non-distributional methods.

**Contributions.** We make three key contributions:

1. We propose **Flow Anchoring** to efficiently and expressively regularize the policy to the dataset behavior.
2. We propose a **noise-conditioned value function defined by the operator** $\mathcal{T}_n^\pi$ to efficiently capture expressive distributional return information.
3. Our proposed algorithm, FAN, achieves high computational efficiency in both training and inference while simultaneously improving offline RL performance.

## 2. Related Work

**Offline RL.** In offline RL, the policy is trained to maximize the expected sum of rewards without further environment interactions. Given a fixed dataset, the primary challenge is to avoid distribution shift caused by value overestimation on out-of-distribution (OOD) actions. Prior work adopts behavior regularization (Wu et al., 2019; Peng et al., 2019; Fujimoto & Gu, 2021; Tarasov et al., 2023a), conservatism (Kumar et al., 2020), in-sample learning (Kostrikov et al., 2022; Garg et al., 2023; Xu et al., 2023), and more (Chen et al., 2023; Nikulin et al., 2023; Sikchi et al., 2024; Lee & Kwon, 2025) to constrain OOD actions to the dataset action support. In this work, FAN applies behavior regularization to constrain the policy to be similar to the dataset behavior.

**Diffusion and Flow Policies in Offline RL.** Recent work in offline RL has increasingly leveraged diffusion and flow policies to address the limitations of unimodal Gaussian policies. By solving the underlying differential equations, these policies provide highly expressive modeling of the offline behavior distribution. Prior work trains diffusion and flow policies using objectives weighted by action values (Ding et al., 2024; Zhang et al., 2025), samples optimal actions through rejection sampling (Hansen-Estruch et al., 2023; He et al., 2024; Mao et al., 2024), or utilizes them for behavior regularization (Chen et al., 2024a;c; Gao et al., 2025; Park et al., 2025c; Lee et al., 2026), and more (Venkatraman et al., 2024; Chen et al., 2024b; Chung et al., 2026). FAN uses flow policies for behavior regularization, but eliminates the computational bottleneck of iterative sampling.

**Distributional Offline RL.** Distributional RL (Engel et al., 2005; Morimura et al., 2010; Bellemare et al., 2017) aims to learn the entire distribution of future returns, rather than just the expected return as in non-distributional approaches. With expressive distributional critics, these methods demonstrate strong theoretical guarantees (Wang et al., 2023; 2024) and empirical performance (Dabney et al., 2018a;b; Farebrother et al., 2024), especially in risk-sensitive settings (Kim et al., 2023; Ma et al., 2025). Prior work in distributional offline RL includes quantile-based (Urpí et al., 2021; Ma et al., 2021), uncertainty-based (Agarwal et al., 2020; Wu et al., 2021), and generative modeling-based (Dong et al., 2026) approaches. However, these methods typically incur significant computational overheads, re-

quiring processes such as updates on multiple quantile samples, ensemble evaluations for variance estimation, or iterative sampling for generative modeling. In contrast, FAN addresses these inefficiencies using a noise-conditioned critic.

## 3. Preliminaries

**Problem Setting.** We consider a Markov Decision Process (MDP) defined as $\mathcal{M} = (\mathcal{S}, \mathcal{A}, r, \mu, P, \gamma)$, where $\mathcal{S}$ is the state space, $\mathcal{A}$ is the $d$-dimensional action space, and $r : \mathcal{S} \times \mathcal{A} \to \mathbb{R}$ is the reward function. The notation $\Delta(\mathcal{X})$ denotes the set of probability distributions over a space $\mathcal{X}$. $\mu \in \Delta(\mathcal{S})$ is the initial state distribution, and $P : \mathcal{S} \times \mathcal{A} \to \Delta(\mathcal{S})$ is the transition dynamics kernel. and $\gamma \in (0, 1)$ is the discount factor. The goal is to learn a policy $\pi : \mathcal{S} \to \Delta(\mathcal{A})$ that maximizes the cumulative discounted return.

The standard action-value function $Q^\pi(s, a)$ in prior work is defined to estimate the expected future return:

$$Q^\pi(s, a) = \mathbb{E}_{\tau \sim P^\pi(\cdot|s_0=s, a_0=a)} \left[ \sum_{t=0}^\infty \gamma^t r(s_t, a_t) \right], \quad (1)$$

where the expectation is taken over trajectories $\tau = (s_0, a_0, r_0, s_1, \dots)$ generated by the dynamics $P$ and policy $\pi$. Offline RL aims to find the optimal policy $\pi^*$ that maximizes the expected return $\mathbb{E}_{s_0 \sim \mu, a_0 \sim \pi(\cdot|s_0)}[Q^\pi(s_0, a_0)]$ using only a fixed dataset $\mathcal{D} = \{\tau^{(i)}\}$ of trajectories.

In this work, instead of using the standard $Q^\pi(s, a)$, we capture the distributional information of the return with our proposed noise-conditioned critic $Q^\pi(s, a, \epsilon)$, where $\epsilon \sim \mathcal{N}(0, I_d)$.

**Behavior-Regularized Actor-Critic (BRAC).** BRAC (Wu et al., 2019; Tarasov et al., 2023a; Park et al., 2025c) is a generalized offline RL framework that achieves state-of-the-art performance by enforcing a constraint between the learned policy $\pi_\omega$ and the dataset behavior policy $\pi_\beta$. Specifically, BRAC incorporates a regularization term $R(\pi_\omega(\cdot|s), \pi_\beta(\cdot|s))$ (e.g., KL divergence or Wasserstein distance between distributions) into the actor-critic updates, resulting in the following coupled objectives:

$$\mathcal{L}_Q(\phi) = \mathbb{E}\left[\left(Q_\phi(s, a) - (r + \gamma q_{\hat\phi}^{\pi_\omega, \pi_\beta}(\cdot|s'))\right)^2\right] =$$
$$\mathbb{E}\left[\left(Q_\phi(s, a) - (r + \gamma(Q_{\hat\phi}(s', a'_\omega) - \alpha_2 R(a'_\omega, a'_\beta)))\right)^2\right],$$
$$\mathcal{L}_\pi(\omega) = \mathbb{E}\left[-Q_\phi(s, a_\omega) + \alpha_1 R(a_\omega, a_\beta)\right], \quad (2)$$

where the expectation is taken over $(s, a, r, s') \sim \mathcal{D}$, $a_\omega \sim \pi_\omega(\cdot|s)$, $a_\beta \sim \pi_\beta(\cdot|s)$, $a'_\omega \sim \pi_\omega(\cdot|s')$, and $a'_\beta \sim \pi_\beta(\cdot|s')$. Here, $\alpha_1, \alpha_2 > 0$ determine the regularization strength, and different choices of $R$ recover different algorithms such as ReBRAC (Tarasov et al., 2023a), FQL (Park et al., 2025c), or the proposed FAN algorithm.

**Flow Matching.** Flow matching (Lipman et al., 2023; Liu et al., 2023; Albergo & Vanden-Eijnden, 2023) is a class of generative modeling that learns the underlying velocity field between a prior distribution and a target distribution. Formally, given a target distribution $p(x) \in \Delta(\mathbb{R}^d)$, a time-dependent velocity field $v(t, x) : [0, 1] \times \mathbb{R}^d \to \mathbb{R}^d$ defines a flow trajectory $\psi(t, x) : [0, 1] \times \mathbb{R}^d \to \mathbb{R}^d$, which serves as the unique solution to the following ordinary differential equation (ODE) (Lee, 2003):

$$\frac{d}{dt}\psi(t, x) = v(t, \psi(t, x)) \quad (3)$$

By satisfying the continuity equation, the velocity field $v$ generates a probability density path $p_t(x)$ that continuously maps the prior noise distribution $p_0(x)$ to the target data distribution $p_1(x)$. Prior work (Lipman et al., 2023) has shown that minimizing the following conditional flow matching (CFM) loss based on the Optimal Transport (OT) path is sufficient for training the underlying vector field.

$$\mathcal{L}_{\text{CFM}}(\theta) = \mathbb{E}_{\substack{x_1 \sim p(x), \\ x_0 \sim \mathcal{N}(0, I), \\ t \sim \text{Unif}([0,1]), \\ x_t = (1-t)x_0 + tx_1}} \left[\|v_\theta(t, x_t) - (x_1 - x_0)\|_2^2\right]$$
$$(4)$$

The learned velocity $v_\theta$ transforms the Gaussian $\mathcal{N}(0, I)$ to the target distribution $p(x)$ through the flow (Eq.(3)). We use Eq.(4) to train our flow policy $v_\theta$, which maps the normal distribution to the offline dataset action distribution.

**Behavior Flow Policy.** In offline RL, the behavior flow policy models the behavior distribution of the offline dataset and is trained with the following objective similar to Eq.(4):

$$\mathcal{L}_{\text{FlowBC}}(\theta) = \mathbb{E}_{\substack{(s,a) \sim \mathcal{D}, \\ \epsilon \sim \mathcal{N}(0, I_d), \\ t \sim \text{Unif}([0,1]), \\ a_t = (1-t)\epsilon + ta}} \left[\|v_\theta(s, t, a_t) - (a - \epsilon)\|_2^2\right]$$
$$(5)$$

Sampling actions from the behavior flow policy $v_\theta$ recovers the dataset behavior, but requires solving Eq.(3) using ODE solvers (e.g., Euler method). To sample actions with higher returns than $v_\theta$, prior work has applied rejection sampling weighted by future return estimates (Park et al., 2026; 2025b; Dong et al., 2026), or trained a separate one-step policy $\pi_\omega$ with behavior regularization (Park et al., 2025c). Rejection sampling requires multiple $v_\theta$ iterations for both training and inference, whereas behavior regularization enables one-step action inference with $\pi_\omega$ trained using the objective:

$$\mathcal{L}_P(\omega) = \mathbb{E}_{\substack{s \sim \mathcal{D}, \\ \epsilon \sim \mathcal{N}(0, I_d), \\ a_\omega = \pi_\omega(s, \epsilon)}} \left[-Q^{\pi_\omega}(s, a_\omega) + \alpha \|a_\omega - a_\theta\|^2\right], \quad (6)$$

where $a_\theta$ is the terminal state of the ODE defined by $v_\theta$ starting from $\epsilon$, $Q^{\pi_\omega}$ is the expected return under the policy $\pi_\omega$, and $\alpha$ is the coefficient for behavior regularization. However, training still requires $v_\theta$ iterations to generate $a_\theta$, which motivates our proposed method, Flow Anchoring.

**Distributional RL.** Instead of expectations, distributional RL focuses on modeling the entire distribution of future returns. Given a policy $\pi$, the discounted return random variable is defined as $Z^\pi = \sum_{t=0}^\infty \gamma^t r(S_t, A_t)$, with values in the range $[z_{\min}, z_{\max}] \triangleq \left[\frac{r_{\min}}{1-\gamma}, \frac{r_{\max}}{1-\gamma}\right]$. Here, $S_t$ and $A_t$ are the state and action random variables at timestep $t$, where their values are determined by trajectories following $\pi$. The conditional return random variable is defined as $Z^\pi(s, a) = r(s, a) + \sum_{h=1}^\infty \gamma^h r(S_h, A_h)$, and the expected return value estimate satisfies $Q^\pi(s, a) = \mathbb{E}[Z^\pi(s, a)]$. The distributional Bellman operator $\mathcal{T}^\pi$ is defined as:

$$\mathcal{T}^\pi Z(s, a) \overset{d}{=} r(s, a) + \gamma Z(S', A'), \tag{7}$$

where $S'$ and $A'$ are random variables following the joint density $P(s'|s, a)\pi(a'|s')$, and $\overset{d}{=}$ denotes equality in distribution. Prior work (Bellemare et al., 2017) has shown that $\mathcal{T}^\pi$ is a $\gamma$-contraction under the $p$-Wasserstein distance, and therefore, repeatedly applying $\mathcal{T}^\pi$ converges to a unique fixed point (Banach, 1922). Our proposed distributional Bellman operator $\mathcal{T}_n^\pi$ (Eq.(9)) also satisfies the conditions of Banach's fixed-point theorem.

**Expectile Loss.** Expectile regression (Newey & Powell, 1987) generalizes standard mean squared error (MSE) loss to an asymmetric form. For a prediction $x$ and a target $\hat{x}$, the expectile loss is defined using the coefficient $\kappa \in (0, 1)$:

$$\mathcal{L}_2^\kappa(\hat{x} - x) = |\kappa - \mathbf{1}((\hat{x} - x) < 0)| (\hat{x} - x)^2. \tag{8}$$

Here, the expectile is the minimizer of Eq.(8), and with fixed $\kappa$, it becomes the $\kappa$-th expectile of the target random variable $\hat{x}$. In distributional RL, approaches such as those by Rowland et al. (2019) and Jullien et al. (2025) model expectiles of the return random variable with Eq.(8). Moreover, non-distributional offline RL with in-sample learning (Kostrikov et al., 2022; Xu et al., 2023; Kim et al., 2026) exploits Eq.(8) to approximate the optimal value $V^*(s) \approx \max_a Q(s, a)$ using the loss $L_V(\psi) = \mathbb{E}_{(s,a)\sim\mathcal{D}}[\mathcal{L}_2^\kappa(Q_{\hat{\theta}}(s, a) - V_\psi(s))]$ with $\kappa \approx 1$. Similarly, we use Eq.(8) to estimate the $\mathrm{ess\,sup}$ in Eq.(9).

# 4. Flow-Anchored Noise-conditioned Q-Learning (FAN)

We now introduce FAN, a behavior-regularized actor-critic method using flow policies and distributional critics. FAN has two key components: (1) the operator $\mathcal{T}_n^\pi$ for critic training, and (2) Flow Anchoring for behavior regularization.

**Main Focus.** Our primary objective is to maximize both performance and efficiency. However, high performance usually incurs higher computational costs. Among various mechanisms, we prioritize the use of *expressive* models to achieve high performance. Specifically, we design the policies to be supported by flow matching and the values to capture return distributions. We aim to maximize computational efficiency within this expressive framework.

**Notations and Function Definitions.** Fix $s \in \mathcal{S}$ and $a \in \mathcal{A}$. Let $\epsilon_p, \epsilon_v \sim \mathcal{N}(0, I_d)$ and $t, \kappa \sim \mathrm{Unif}([0, 1])$, where we mark random variables in gray. A stochastic policy is a measurable map $\pi : \mathcal{S} \times \mathbb{R}^d \to \mathcal{A}$, and the sampled action is $a_\pi := \pi(s, \epsilon_p)$. For behavior regularization, we define a behavior flow policy as a measurable map $v : \mathcal{S} \times [0, 1] \times \mathcal{A} \to \mathcal{A}$, where $v_\beta := v(s, t, a_t)$ models the velocity field associated with the offline behavior action distribution using $a_t := (1 - t)\epsilon_p + t a$. Let $\mathcal{Q}$ be the space of bounded, measurable functions $\mathcal{S} \times \mathcal{A} \times \mathbb{R}^d \to \mathbb{R}$, and fix $Q \in \mathcal{Q}$. Then $Q_n := Q(s, a, \epsilon_v)$ is a random variable, and we define $Q^\pi \in \mathcal{Q}$ as the unique fixed point of $\mathcal{T}_n^\pi$ (Eq. (9)) by Theorem 4.1. Finally, we define the $\kappa$-th expectile of $Q_n$ as $Z_\kappa^Q := \arg\min_{q \in \mathbb{R}} \mathbb{E}_{\epsilon_v \sim \mathcal{N}(0, I_d)} \left[\mathcal{L}_2^\kappa(Q(s, a, \epsilon_v) - q)\right]$.

**Value Networks.** We train two function approximators: $Q_\phi(s, a, \epsilon)$ to model $Q^\pi(s, a, \epsilon)$, and $Z_\psi(s, a)$ for the upper expectile of $Q_\phi(s, a)$, which is $Z_{\kappa \approx 1}^{Q_\phi}$. By Theorem 4.2, $\lim_{\kappa \to 1^-} Z_\psi(s, a) = \mathrm{ess\,sup}_{\epsilon \sim \mathcal{N}(0, I_d)} Q_\phi(s, a, \epsilon)$.

**Policy Networks.** We use two policy neural networks: $\pi_\omega(s, \epsilon)$ for modeling $\pi$, and $v_\theta(s, t, a_t)$ for modeling $v$.

## 4.1. Actor-Critic Training

**Motivation.** One of the major computational bottlenecks in distributional critic training is that it requires analyzing multiple samples (e.g., quantiles). *However, should we always rely on multiple samples to use the distributional information of future returns?* As one solution, we propose to use noise vectors instead of quantiles, setting the distributional critic training remain valid even with a single sample. Specifically, with $\epsilon' \sim \mathcal{N}(0, I_d)$, we define the following distributional operator on $Q(s, a, \epsilon')$:

$$\mathcal{T}_n^\pi Q(s, a, \epsilon') \overset{d}{:=} r + \gamma \operatorname*{ess\,sup}_{\epsilon \sim \mathcal{N}(0, I_d)} Q(s', \pi(s', \epsilon'), \epsilon). \tag{9}$$

For simplicity, we only consider deterministic transitions and rewards, meaning that the reward $r$ and the next state $s'$ are fixed given $(s, a)$. The convergence of the operator is guaranteed by Theorem 4.1, and therefore, iteratively applying $\mathcal{T}_n^\pi$ to any $Q \in \mathcal{Q}$ converges to $Q^\pi$. The motivation for using the $\mathrm{ess\,sup}$ is to preserve the greedy, max-based action selection principle underlying classical Q-learning (Watkins & Dayan, 1992), while extending it to noise-conditioned return distributions. We refer to Appendix A for detailed explanations and theoretical benefits of $\mathcal{T}_n^\pi$.

**(1) Noise-conditioned Critic Update.** Direct Monte Carlo sampling for estimating the essential supremum in Eq.(9) requires multiple noise samples. Moreover, a max operation on these samples can also increase value overestimation. Therefore, we propose the following Temporal Difference

(TD) learning objective for the noise-conditioned critic $Q_\phi$:

$$\mathcal{L}_Q(\phi) = \mathbb{E}\left[(Q_\phi(s,a,\epsilon') - (r + \gamma\, q_\psi^{\pi_\omega, v_\theta}(s', \epsilon'))^2)\right], \quad (10)$$

where the expectation is taken over $(s,a,r,s') \sim \mathcal{D}$ and $\epsilon' \sim \mathcal{N}(0, I_d)$. $q^{\pi_\omega, v_\theta}(s', \epsilon') := Z_\psi(s', \pi_\omega(s', \epsilon')) - \alpha_2 \hat{z}$ is the behavior regularized critic value defined in Eq.(15).

**(2) Upper Expectile Regression.** We train $Z_\psi(s,a)$ to model the ess sup in $\mathcal{T}_n^\pi$ (Eq.(9)), using only the state-action data pairs in the offline dataset:

$$\mathcal{L}_Z(\psi) = \mathbb{E}_{\substack{(s,a)\sim\mathcal{D},\\ \epsilon\sim\mathcal{N}(0,I_d)}} \left[L_2^\kappa(Q_{\hat{\phi}}(s,a,\epsilon) - Z_\psi(s,a))\right]. \quad (11)$$

To make $Z_\psi$ model the upper expectile, we fix $\kappa = 0.9$ for all experiments, which differs from prior distributional approaches that train for all possible $\kappa \sim \text{Unif}([0,1])$.

**(3) Value Maximization.** The one-step policy $\pi_\omega$ is trained to maximize the estimated future return by minimizing:

$$\mathcal{L}_P(\omega) = \mathbb{E}_{\substack{s\sim\mathcal{D},\\ \epsilon,\epsilon'\sim\mathcal{N}(0,I_d),\\ a_\omega=\pi_\omega(s,\epsilon)}} \left[-Q_\phi(s, a_\omega, \epsilon') - Z_\psi(s, a_\omega)\right]. \quad (12)$$

With Eq.(12), the actor seeks the highest possible return using both $Q_\phi$ and $Z_\psi$.

### 4.2. Behavior Regularization

**Motivation.** Prior work on flow policy behavior regularization requires dataset actions sampled through ODE solving. *However, is exact behavior sampling necessary for regularization?* Instead of using exact action samples, we propose "Flow Anchoring", which regularizes both the policy and value networks without ODE solutions. Since we regularize both the actor and the critic, FAN falls into the category of behavior-regularized actor-critic (Wu et al., 2019).

**(1) Behavior Flow Policy.** As in prior work (Park et al., 2025c; Dong et al., 2026), we clone the dataset behavior using flow matching. Specifically, the behavior policy models the vector field mapping the Gaussian distribution to the state-conditional action distribution of the offline dataset:

$$\mathcal{L}_F(\theta) = \mathbb{E}_{\substack{(s,a)\sim\mathcal{D},\\ t\sim\text{Unif}([0,1]),\\ \epsilon\sim\mathcal{N}(0,I_d),\\ a_t=(1-t)\epsilon+ta}} \left[\|v_\theta(s, t, a_t) - (a - \epsilon)\|_2^2\right]. \quad (13)$$

**(2) Actor Flow Anchoring.** Behavior regularization with Eq.(6) increases training computation due to iterative flow sampling. In contrast, we propose Eq.(14) which regularizes both the actor and critic separately, without flow iteration. This provides an efficient and effective action regularization with the underlying flow of the offline dataset actions:

$$\mathcal{L}_B(\omega) = \mathbb{E}\left[\|(\pi_\omega(s,\epsilon) - \epsilon) - v_\theta(s, t, a_{t,\omega})\|_2^2\right]. \quad (14)$$

---

**Algorithm 1** FAN

**Input:** Dataset $\mathcal{D}$, one-step policy $\pi_\omega$, behavior flow policy $v_\theta$, noise-conditioned critic $Q_\phi$, critic upper expectile estimator $Z_\psi$, $\kappa = 0.9$, $\tau = 0.995$, behavior regularization coefficients $\alpha_1, \alpha_2$.

**while** *not converged* **do**
  Sample batch $B = \{(s,a,r,s')\} \sim \mathcal{D}$
  `ValueUpdate`$(B)$, `PolicyUpdate`$(B)$
  $\hat{\phi} \leftarrow \tau\phi + (1-\tau)\hat{\phi}$
**return** One-step policy $\pi_\omega$

**Function** `ValueUpdate`$(B)$:
  $\epsilon', \epsilon_1, \epsilon_2, \epsilon \sim \mathcal{N}(0, I_d), \quad t \sim \text{Unif}([0,1])$
  $a'_\omega \leftarrow \pi_\omega(s', \epsilon'), \quad a'_{t,\omega} \leftarrow (1-t)\epsilon' + ta'_\omega$
  $z \leftarrow Z_\psi(s', a'_\omega), \quad \hat{z} \leftarrow \|(a'_\omega - \epsilon') - v_\theta(s', t, a'_{t,\omega})\|_2^2$

  ▷ TD update with Critic Flow Anchoring
  $L_Q(\phi) \leftarrow \mathbb{E}[(Q_\phi(s,a,\epsilon') - (r + \gamma(z - \alpha_2\hat{z})))^2]$
  ▷ Upper Expectile Regression
  $L_Z(\psi) \leftarrow \mathbb{E}[L_2^\kappa(Q_{\hat{\phi}}(s,a,\epsilon) - Z_\psi(s,a))]$

  Update $\phi, \psi$ to minimize $L_Q + L_Z$

**Function** `PolicyUpdate`$(B)$:
  $\epsilon_1, \epsilon_2, \epsilon_3 \sim \mathcal{N}(0, I_d), \quad t_1, t_2 \sim \text{Unif}([0,1])$
  $a_t \leftarrow (1-t_1)\epsilon_1 + t_1 a$
  $a_\omega \leftarrow \pi_\omega(s, \epsilon_2), \quad a_{t,\omega} \leftarrow (1-t_2)\epsilon_2 + t_2 a_\omega$

  ▷ Behavior Flow Matching
  $L_F(\theta) \leftarrow \mathbb{E}[\|v_\theta(s, t_1, a_t) - (a - \epsilon_1)\|_2^2]$
  ▷ Actor Flow Anchoring
  $L_B(\omega) \leftarrow \mathbb{E}[\|(a_\omega - \epsilon_2) - v_\theta(s, t_2, a_{t,\omega})\|_2^2]$
  ▷ Value Maximization
  $L_P(\omega) \leftarrow \mathbb{E}[-Q_\phi(s, a_\omega, \epsilon_3) - Z_\psi(s, a_\omega)]$

  Update $\theta, \omega$ to minimize $L_F + \alpha_1 L_B + L_P$

---

The expectation is taken over $t \sim \text{Unif}([0,1])$, $\epsilon \sim \mathcal{N}(0, I_d)$, and $s \sim \mathcal{D}$, with $a_{t,\omega} = (1-t)\epsilon + t\pi_\omega(s,\epsilon)$.

**(3) Critic Flow Anchoring.** We also incorporate behavior regularization into Eq.(10) with coefficient $\alpha_2$:

$$q_\psi^{\pi_\omega, v_\theta}(s', \epsilon') := Z_\psi(s', \pi_\omega(s', \epsilon'))$$
$$- \alpha_2 \mathbb{E}_{t\sim\text{Unif}([0,1])}\left[\|(\pi_\omega(s', \epsilon') - \epsilon') - v_\theta(s', t, a'_{t,\omega})\|_2^2\right], \quad (15)$$

where $a'_{t,\omega} = (1-t)\epsilon' + t\pi_\omega(s', \epsilon')$.

### 4.3. Theoretical Guarantees

Although the motivations are from computational efficiency, we show that the details in FAN are actually solid in theory.

**(1) Convergence of the operator $\mathcal{T}_n^\pi$ (Eq.(9)).** As the

standard distributional operator $\mathcal{T}^\pi$ (Eq.(7)) guarantees convergence in the $p$-Wasserstein metric ($W_p$) (Bellemare et al., 2017), we establish convergence in $\infty$-Wasserstein metric ($W_\infty$). Specifically, we prove that $\mathcal{T}^\pi_n$ is a $\gamma$-contraction in the supremum metric $d_\infty$ (Definition B.1), a condition that strictly implies distributional convergence under $W_\infty$.

> **Theorem 4.1** (**Convergence of $\mathcal{T}^\pi_n$**). *The proposed operator $\mathcal{T}^\pi_n$ is a $\gamma$-contraction on $(\mathcal{Q}, d_\infty)$ (Definition B.1), and therefore, iterating $\mathcal{T}^\pi_n$ from any $Q \in \mathcal{Q}$ converges to a unique fixed point $Q^\pi$.*

*Proof.* Please refer to the proof in Appendix B.1. Therefore, for any $s \in \mathcal{S}$, $a \in \mathcal{A}$, $\epsilon' \in \mathbb{R}^d$, and $Q \in \mathcal{Q}$, $Q(s, a, \epsilon')$ converges to $Q^\pi(s, a, \epsilon')$ if we iterate $\mathcal{T}^\pi_n$ over this value. □

**(2) Upper Expectile and the Essential Supremum.** We now show why our TD learning objective (Eq.(10)) recovers the return distribution converged through $\mathcal{T}^\pi_n$ (Eq.(9)).

> **Theorem 4.2** (**Upper Expectile Converges to the Essential Supremum**). *Let $s \in \mathcal{S}$, $a \in \mathcal{A}$, $\epsilon \sim \mathcal{N}(0, I_d)$, and $Q \in \mathcal{Q}$. For any $\kappa \in [\frac{1}{2}, 1)$, $Z_\kappa := \arg\min_{q \in \mathbb{R}} \mathbb{E}_\epsilon[\mathcal{L}^\kappa_2(Q(s, a, \epsilon) - q)]$ is bounded by:*
>
> $$Z_{1/2} \le Z_\kappa \le \lim_{\kappa \to 1^-} Z_\kappa = \operatorname{ess\,sup}_\epsilon Q(s, a, \epsilon). \tag{16}$$

*Proof.* Please refer to the proof stated in Appendix B.2. This implies that the upper expectile $Z_\psi$ trained through Eq.(11) with $\kappa \approx 1$ converges to $\operatorname{ess\,sup} Q_\phi$. □

**(3) Validity of Behavior Regularization.** We show that minimizing $\mathcal{L}_B$ (Eq.(14)) controls the deviation between distributions induced by the one-step policy $\pi_\omega$ and the behavior policy $v_\theta$ modeling the offline dataset behavior.

> **Theorem 4.3** (**Flow Anchoring is a Valid Behavior Regularization**). *Let $\mu_\omega(\cdot|s)$ and $\mu_\theta(\cdot|s)$ be the probability distributions induced by the policy $\pi_\omega$ and the behavior flow $v_\theta$ respectively (Definition B.5). If $v_\theta$ satisfies Lipschitzness (Assumption B.6), the following holds for all $s \in \mathcal{S}$:*
>
> $$\mathbb{E}_{s \sim \mathcal{D}}\left[W_2^2(\mu_\omega(\cdot|s), \mu_\theta(\cdot|s))\right] \le e^{2L} \mathcal{L}_B(\omega), \tag{17}$$
>
> *where $W_2$ is the Wasserstein-2 distance and $L$ is the Lipschitz constant.*

*Proof.* We provide the complete derivation in Appendix B.3. The equality holds when $\mu_\omega(\cdot|s) = \mu_\theta(\cdot|s)$ and all flow trajectories of the vector field $v_\theta$ are straight. We note that our behavior model $v_\theta$ is parameterized by standard neural networks which are Lipschitz, also with Lipschitz-continuous activation functions (e.g., GeLU). Since the composition of Lipschitz functions is Lipschitz, Assumption B.6 is always satisfied. Consequently, minimizing $\mathcal{L}_B$ (Eq. (14)) directly minimizes the upper bound on the Wasserstein-2 distance between the distributions induced by the training policy $\pi_\omega$ and the behavior flow policy $v_\theta$. □

## 5. Experiments

In this section, we demonstrate that FAN effectively translates theoretical insights into practice. The goal is to observe whether FAN achieves state-of-the-art performance on offline RL benchmarks, while offering high computational efficiency in both training and inference.

**Baselines.** We benchmark FAN against highly efficient non-distributional algorithms, as well as high-performing distributional methods. Therefore, we select **ReBRAC** (Tarasov et al., 2023a), **IDQL** (Hansen-Estruch et al., 2023), and **FQL** (Park et al., 2025c) as non-distributional baselines, and **IQN** (Dabney et al., 2018a), **CODAC** (Ma et al., 2021), and **Value Flows** (Dong et al., 2026) as distributional baselines. With the non-distributional approaches, we mainly focus on comparing the final performance, whereas with the distributional approaches, we mainly compare computational efficiency. Please refer to Appendix C for more baseline details.

### 5.1. Offline RL Task Performance

Now, we report how the policy trained with FAN performs on offline RL benchmarks.

**Benchmarks.** We present results on the standard offline RL benchmarks for robotics locomotion and manipulation. Specifically, we evaluate on **4** antmaze and **12** adroit tasks from D4RL (Fu et al., 2020), and also **25** state-based and **4** pixel-based tasks from OGBench (Park et al., 2025a).

**Settings.** For OGBench tasks, following the official evaluation scheme (Park et al., 2025a), we train for 1M gradient steps for state-based tasks and 500K steps for pixel-based tasks, and report the average success rates across the last three evaluation epochs (i.e., at 100K intervals). For D4RL tasks, we train for 500k gradient steps and report the performance at the last epoch following Tarasov et al. (2023b). For the baselines, we source the best results reported in prior work (Park et al., 2025c; Dong et al., 2026) where tasks overlap, or tune baselines with training budgets similar to FAN if no prior results exist. Appendix C has more experimental details.

*Table 1.* **Offline Results** including normalized returns (D4RL) and success rates (OGBench singletask). The results are bolded if they are within the 95% range of the best final performance in each task. We used 8 seeds for training D4RL and OGBench state-based tasks, and 4 seeds for OGBench pixel-based tasks. The full results are at Table 7, with hyperparameters stated in Tables 3 and 4.

| | | NON-DISTRIBUTIONAL | | | DISTRIBUTIONAL | | | |
| BENCHMARK | TASK TYPES | REBRAC | IDQL | FQL | IQN | CODAC | VF | FAN |
|---|---|---|---|---|---|---|---|---|
| D4RL | ANTMAZE (4 TASKS) | 73 | 75 | **79**±8 | 46±4 | 46±3 | 17±4 | **76**±4 |
| | ADROIT (12 TASKS) | **59** | 52±4 | 52±3 | 50±3 | 52±1 | 50±2 | 53±4 |
| OGBENCH | ANTSOCCER-ARENA-NAVIGATE (5 TASKS) | 16±1 | 33±6 | **60**±2 | 24±7 | 33±14 | 27±7 | **60**±8 |
| | PUZZLE-3X3-PLAY (5 TASKS) | 22±2 | 19±1 | 30±4 | 15±1 | 20±5 | 87±13 | **100**±1 |
| | PUZZLE-4X4-PLAY (5 TASKS) | 14±3 | 25±8 | 17±5 | 27±4 | 20±18 | 27±4 | **42**±10 |
| | CUBE-DOUBLE-PLAY (5 TASKS) | 15±6 | 14±5 | 29±6 | 42±8 | 61±6 | **69**±4 | 46±11 |
| | SCENE-PLAY (5 TASKS) | 45±5 | 30±4 | 56±2 | 40±1 | 55±1 | 59±4 | **58**±1 |
| | VISUAL LOCOMOTION (2 TASKS) | 28±11 | 44±4 | 17±2 | 32±4 | **49**±2 | 44±4 | **49**±4 |
| | VISUAL MANIPULATION (2 TASKS) | 16±4 | 8±11 | 28±5 | 6±3 | 2±1 | 30±4 | **33**±16 |

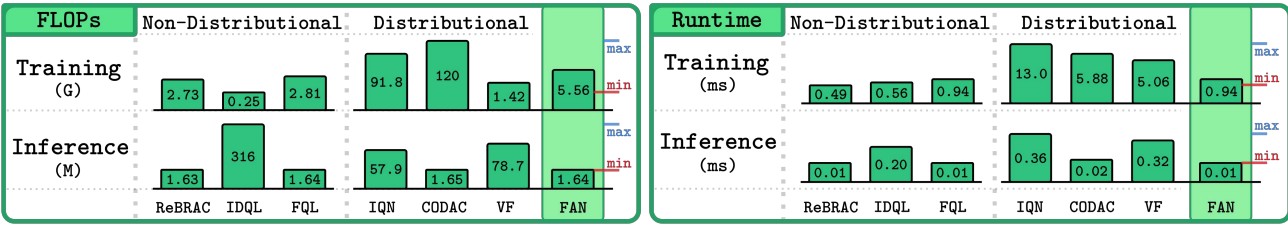

*Figure 3.* **The Number of FLOPs and the Wall-clock Compute Time** per function call for `cube-double-play`.

**Results.** Table 1 presents results on the performance. FAN achieves state-of-the-art performance in 7 out of 9 task environments, where we define state-of-the-art as achieving at least 95% of the best task performance. Specifically, FAN outperforms non-distributional approaches in most OGBench tasks, especially for the tasks dealing with complex manipulation (e.g., `puzzle`, `cube`). Also, FAN surpasses distributional approaches on average while maintaining higher computational efficiency.

### 5.2. Computational Efficiency

We evaluate computational efficiency using both the number of floating-point operations (FLOPs) and wall-clock runtime. To quantify computational costs for both training and inference, we measure these metrics for a single training update and a single action-sampling call. All measurements are performed on a single NVIDIA RTX 6000 GPU using JAX/XLA with a batch size of 256. For the baselines, we standardized by using 16 quantiles, 16 action candidates for rejection sampling, and 10 flow steps if needed.

**Floating Point Operations (FLOPs).** We measure FLOPs using XLA static cost analysis (OpenXLA Team, 2017) in JAX (Bradbury et al., 2018). Concretely, we JIT-compile each measured function into an XLA executable and report the compiler-estimated FLOPs for one execution of the compiled graph. For training, we measure FLOPs of the actor-critic updates, which include the forward/backward pass, optimizer updates, and target-network updates. For inference, we measure FLOPs of a single action sampling.

**Wall-Clock Compute Time.** We measure wall-clock runtime for both training and inference, excluding compilation overhead before measurements. We report the mean runtime per call calculated over 50 runs.

**Results.** Figure 3 summarizes the two metrics. For training, CODAC and IQN incur substantially higher costs because they use multiple samples to learn distributional value functions. Value Flows, which is based on flow integration and Jacobian–vector products, exhibits low training FLOPs due to efficient code compilation but actually requires high runtime. Compared to these methods, FAN results in approximately a 5–14× faster runtime during training, demonstrating the highest computational efficiency among the distributional approaches. Moreover, for inference, FAN shows the best computational efficiency over all baseline methods, in terms of both the number of FLOPs and runtime.

### 5.3. Ablation Studies

We now analyze the role of each component in FAN. First, we compare Flow Anchoring with prior behavior regularization techniques, and second, we compare $\mathcal{T}_n^\pi$ to the standard expected action value functions. Moreover, we investigate how FAN performs in the offline-to-online setting. Further ablation studies include experiments on the effect of $\kappa$, the effect of using both $Z_\psi$ and $Q_\phi$, and the effect of using multiple noise samples for the value estimation. The results were collected from training on the five default tasks of OGBench (`antsoccer`, `scene`, `cube-double`, `puzzle-3x3`, and `puzzle-4x4`), following the hyperparameters stated

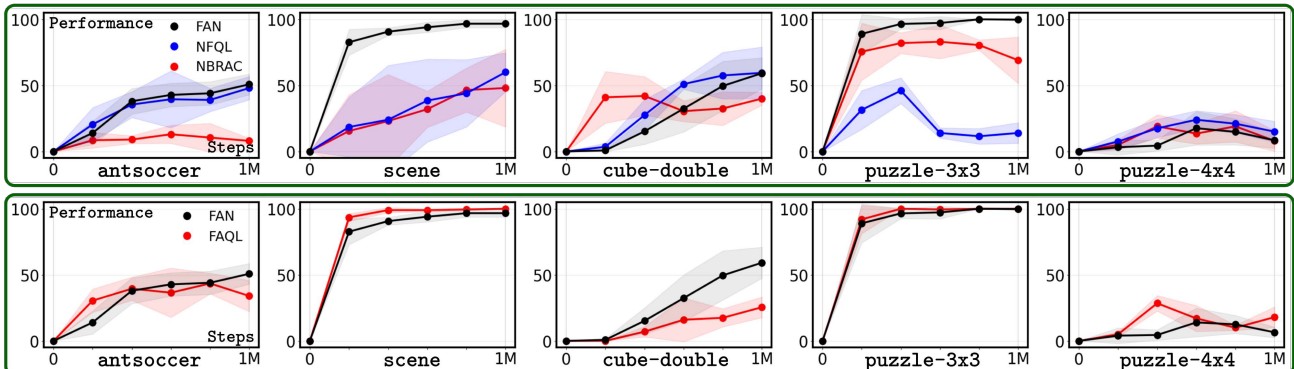

*Figure 4.* **Ablation Studies on Flow Anchoring and $\mathcal{T}_n^\pi$.** *(Up)* NBRAC vs. NFQL vs. FAN to verify the effect of Flow Anchoring. *(Down)* FAQL vs. FAN to verify the effect of $\mathcal{T}_n^\pi$. The black line (FAN) performs the best on average, compared to all other combinations.

*Table 2.* **Offline-to-Online Results** including normalized returns (D4RL) and success rates (OGBench `singletask-v0` defaults). The results are collected over 8 seeds and the numbers are bolded if they are above or equal to 95% of the best performance.

| | | NON-DISTRIBUTIONAL | | | DISTRIBUTIONAL | | |
|---|---|---|---|---|---|---|---|
| BENCHMARK | TASK | ReBRAC | IDQL | FQL | IQN | VALUE FLOWS | FAN |
| OGBENCH | ANTSOCCER-MEDIUM-NAVIGATE | $0\pm0 \to 0\pm0$ | $26\pm15 \to 39\pm10$ | $28\pm8 \to \mathbf{86}\pm5$ | $28\pm8 \to 34\pm4$ | $22\pm3 \to 0\pm0$ | $52\pm8 \to 68\pm9$ |
| | SCENE-PLAY | $55\pm10 \to \mathbf{100}\pm0$ | $0\pm1 \to 60\pm39$ | $82\pm11 \to \mathbf{100}\pm1$ | $0\pm0 \to 0\pm0$ | $92\pm23 \to \mathbf{100}\pm0$ | $96\pm2 \to \mathbf{100}\pm0$ |
| | CUBE-DOUBLE-PLAY | $6\pm5 \to 28\pm28$ | $12\pm3 \to 41\pm2$ | $40\pm11 \to 92\pm3$ | $29\pm4 \to 42\pm7$ | $65\pm7 \to 79\pm6$ | $59\pm13 \to \mathbf{98}\pm2$ |
| | PUZZLE-3X3-PLAY | $90\pm5 \to \mathbf{100}\pm0$ | $6\pm7 \to 0\pm0$ | $75\pm11 \to 73\pm38$ | $58\pm42 \to 84\pm7$ | $2\pm3 \to 0\pm0$ | $99\pm1 \to \mathbf{100}\pm0$ |
| | PUZZLE-4X4-PLAY | $8\pm4 \to 14\pm35$ | $23\pm2 \to 19\pm12$ | $8\pm3 \to 38\pm52$ | $22\pm2 \to 6\pm1$ | $14\pm3 \to 51\pm12$ | $17\pm7 \to \mathbf{100}\pm1$ |

in Tables 5 and 6.

**Why Flow Anchoring?** Besides its computational efficiency, we investigate how our behavior regularization affects final performance. For this, we fix training the noise-conditioned critic with $\mathcal{T}_n^\pi$ and compare three different behavior regularization techniques: **NBRAC** using actor-critic standard behavior cloning (BC) from ReBRAC (Tarasov et al., 2023a), **NFQL** using actor flow BC from FQL (Park et al., 2025c), and **FAN** using actor-critic Flow Anchoring. The upper part of Figure 4 shows that Flow Anchoring leads to better performance (or performance within 95% of the best) in 4 out of 5 tasks. Therefore, we conclude that Flow Anchoring is the behavior regularization technique that best suits $\mathcal{T}_n^\pi$, in terms of both task performance and efficiency.

**Why $\mathcal{T}_n^\pi$?** We also investigate how training for $\mathcal{T}_n^\pi$ performs with Flow Anchoring. For this, we compare FAN with FAQL, which is a variant of FQL using the standard non-distributional Bellman operator but with Flow Anchoring. The lower part of Figure 4 shows that $\mathcal{T}_n^\pi$ leads to better performance (or performance within 95% of the best) on 4 out of 5 tasks. Hence, we conclude that $\mathcal{T}_n^\pi$ helps improve performance when used with Flow Anchoring.

**Offline-to-Online.** We further evaluate how FAN performs during online fine-tuning. For this, we conduct an additional 1M steps of training with environment interactions after the initial 1M step offline training. Specifically, we lower the $\alpha_1, \alpha_2$ values in the online phase, relaxing constraints to allow for broad exploration. According to Table 2, FAN achieves state-of-the-art performance on 4 out of 5 tasks,

and therefore, we conclude that FAN also performs well in offline-to-online settings.

**Sensitivity to $\kappa$.** We analyze how varying $\kappa$ affects the final performance of the policy. We present evaluations on OGBench `antsoccer-arena-navigate-task1` ($\alpha_1 = 10, \alpha_2 = 0.1$) and `puzzle-4x4-play-task1` ($\alpha_1 = 100, \alpha_2 = 3$), with $\kappa \in \{0.5, 0.7, 0.9, 0.99\}$.

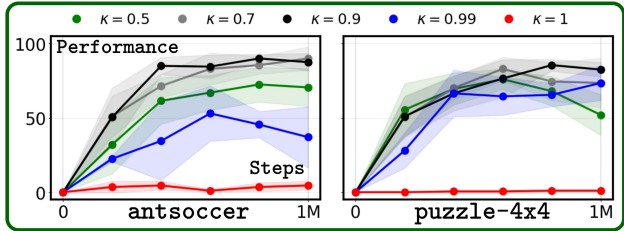

*Figure 5.* **Ablation Study on Sensitivity to $\kappa$.** The black line ($\kappa = 0.9$) empirically achieves the best average performance.

In Figure 5, setting $\kappa = 0.9$ yields the best performance on these two tasks. Performance improves as $\kappa$ increases from 0.5 to 0.9. However, setting $\kappa$ too close to 1 (e.g., $\kappa = 0.99$ or 1) leads to performance degradation. Therefore, we fix $\kappa = 0.9$ for the main experiments, reducing the hyperparameter search space to only $\alpha_1$ and $\alpha_2$.

**Why Maximize both $Z_\psi$ and $Q_\phi$?** For policy training, we evaluated how performance varies across three configurations: (1) maximizing both $Z_\psi$ and $Q_\phi$, (2) maximizing only $Q_\phi$, and (3) maximizing only $Z_\psi$. We conducted evaluations on five tasks within the OGBench

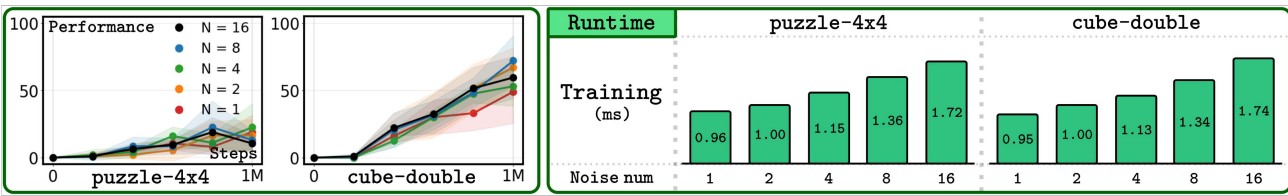

*Figure 6.* **Ablation Study on Value Maximization in Policy Training.** The black line (maximizing both $Z_\psi$ and $Q_\phi$) empirically achieves the best average performance compared to maximizing either component individually.

*Figure 7.* **Ablation Study on Increased Number of Noise Samples for Value Training.** *(Left)* Performance curves with varying numbers of noise samples. *(Right)* Runtime comparison with varying numbers of noise samples.

`antsoccer-arena-navigate` environment, setting $\alpha_1 = 10$ and $\alpha_2 = 0.1$. Figure 6 demonstrates that maximizing both $Z_\psi$ and $Q_\phi$ yields superior performance, justifying our design choice in Eq.(12).

**More Noise Samples for Training $Q_\phi$?** Although we utilize a single noise sample per $Q_\phi$ update, we investigate how increasing the number of noise samples (analogous to using multiple quantiles) affects policy performance. To this end, we conducted evaluations on the default tasks of the OGBench `puzzle-4x4-play` ($\alpha_1 = 100, \alpha_2 = 3$) and `cube-double-play` ($\alpha_1 = 100, \alpha_2 = 0$) environments.

Figure 7 shows that performance does not significantly improve even if we increase the number of noise samples for training the value function. While we observe that the training runtime increases sub-linearly, the added computational cost does not yield proportional performance gains. Consequently, we adopt a single noise sample for training $Q_\phi$.

## 6. Conclusion

In this work, we aimed to achieve state-of-the-art offline RL performance while maximizing computational efficiency. Recognizing that expressive function approximators are crucial for high performance, we investigated how to efficiently employ generative modeling and distributional return information. Our proposed method, FAN, addresses this challenge by leveraging Flow Anchoring and the operator $\mathcal{T}_n^\pi$, both of which are theoretically grounded. Empirical results demonstrate that FAN achieves superior performance and efficiency, while ablation studies validate the individual contributions of our design choices. Finally, we highlighted FAN's strong capabilities in offline-to-online adaptation.

We believe FAN opens several avenues for future work.

First, the concept of Flow Anchoring holds promise for online RL settings with flow policies. Since Flow Anchoring does not directly sample dataset actions, it is effectively complemented by environment interaction, as observed in our offline-to-online experiments. Therefore, applying it to off-policy online RL could yield benefits. Second, beyond efficiency, future research could focus on leveraging $\mathcal{T}_n^\pi$ to maximize task performance. For example, extending its application to model-based RL, risk-sensitive tasks, or goal-conditioned settings represents a promising direction.

## Acknowledgements

This research was supported in part by NSF grant SHF-2505085, and by the W.A."Tex" Moncrief Chair of Computing at the University of Texas at Austin.

## Impact Statement

This paper presents work whose goal is to advance the field of Machine Learning, specifically by improving the computational efficiency of offline reinforcement learning. By reducing the floating point operations (FLOPs) and runtime required for training and inference, our method contributes to energy-efficient AI and facilitates deploying capable policies on resource-constrained robotic hardware. While widespread deployment of autonomous agents carries inherent societal implications—ranging from safety challenges to economic impacts—these risks are intrinsic to reinforcement learning as a whole. Because our contribution focuses strictly on algorithmic efficiency rather than enabling disruptive capabilities, we do not foresee negative consequences unique to this method that require emphasis beyond standard safety considerations.

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

# Appendix

# A. Details on the operator $\mathcal{T}_n^\pi$

Recall that the operator for the noise-conditioned critic is defined as

$$\mathcal{T}_n^\pi Q(s, a, \epsilon') \overset{d}{=} r + \gamma \, \text{ess sup}_{\epsilon \sim \mathcal{N}(0, I_d)} \, Q(s', \pi(s', \epsilon'), \epsilon), \quad \epsilon' \sim \mathcal{N}(0, I_d). \tag{18}$$

This section introduces the measure-theoretic objects used by the operator $\mathcal{T}_n^\pi$ and highlights three theoretical motivations.

## A.1. Measure-Theoretic Notation for $\mathcal{T}_n^\pi$

**$\sigma$-algebras.** A $\sigma$-*algebra* on a set $\Omega$ is a collection $\mathcal{F} \subseteq 2^\Omega$ satisfying:

1. $\Omega \in \mathcal{F}$,

2. if $A \in \mathcal{F}$, then its complement $A^c := \Omega \setminus A$ also belongs to $\mathcal{F}$,

3. if $\{A_i\}_{i=1}^\infty \subseteq \mathcal{F}$, then the countable union $\bigcup_{i=1}^\infty A_i$ belongs to $\mathcal{F}$.

Elements of a $\sigma$-algebra are called *measurable sets* or *events*. These closure properties ensure that probabilistic statements remain well defined under standard set operations and under limiting constructions arising from countable unions and intersections. Given a set $\Omega$, the smallest $\sigma$-algebra containing a collection $\mathcal{C} \subseteq 2^\Omega$ is the *$\sigma$-algebra generated by $\mathcal{C}$*.

**Topological spaces, Borel sets, and Borel measures.** Let $\mathcal{X}$ be a topological space (e.g., $\mathbb{R}$ or $\mathbb{R}^d$ with the usual Euclidean topology). The *Borel $\sigma$-algebra* on $\mathcal{X}$, denoted $\mathcal{B}(\mathcal{X})$, is the smallest $\sigma$-algebra containing all open subsets of $\mathcal{X}$; its elements are called *Borel sets*. A *Borel probability measure* on $\mathcal{X}$ is a function $\mu : \mathcal{B}(\mathcal{X}) \to [0, 1]$ satisfying: (i) $\mu(\mathcal{X}) = 1$, (ii) $\mu(A) \geq 0$ for all $A \in \mathcal{B}(\mathcal{X})$, and (iii) for any pairwise disjoint collection $\{A_i\}_{i=1}^\infty \subseteq \mathcal{B}(\mathcal{X})$, $\mu\left(\bigcup_{i=1}^\infty A_i\right) = \sum_{i=1}^\infty \mu(A_i)$. We denote by $\mathscr{P}(\mathcal{X})$ the set of all Borel probability measures on $\mathcal{X}$.

**Probability space and random variables.** A *probability space* is a triple $(\Omega, \mathcal{F}, \mathbb{P})$, where $\Omega$ is the sample space, $\mathcal{F}$ is a $\sigma$-algebra of measurable events on $\Omega$, and $\mathbb{P}$ is a probability measure on $(\Omega, \mathcal{F})$. A real-valued *random variable* is a measurable map

$$X : (\Omega, \mathcal{F}) \to (\mathbb{R}, \mathcal{B}(\mathbb{R})),$$

where $\mathcal{B}(\mathbb{R})$ is the Borel $\sigma$-algebra on $\mathbb{R}$. The *distribution* (law) of $X$ is the pushforward measure $\mathcal{L}(X) := X_\# \mathbb{P} \in \mathscr{P}(\mathbb{R})$.

**Pushforward measure (#).** Let $\mathscr{P}(\mathbb{R})$ denote the set of Borel probability measures on $\mathbb{R}$. For a measurable function $f : \mathcal{X} \to \mathbb{R}$ and a probability measure $\mu$ on $\mathcal{X}$, the *pushforward* measure $f_\# \mu \in \mathscr{P}(\mathbb{R})$ is defined by

$$(f_\# \mu)(A) := \mu\big(f^{-1}(A)\big), \qquad \forall A \in \mathcal{B}(\mathbb{R}). \tag{19}$$

Equivalently, if $X \sim \mu$, then $f(X) \sim f_\# \mu$.

**Dirac measure.** For $x \in \mathbb{R}$, the Dirac measure $\delta_x \in \mathscr{P}(\mathbb{R})$ is defined by $\delta_x(A) = \mathbf{1}\{x \in A\}$ for all Borel sets $A \subset \mathbb{R}$.

**Essential supremum.** Let $X : (\Omega, \mathcal{F}) \to (\mathbb{R}, \mathcal{B}(\mathbb{R}))$ be a random variable. Its *essential supremum* (w.r.t. $\mathbb{P}$) is

$$\text{ess sup } X := \inf\big\{c \in \mathbb{R} : \mathbb{P}(X > c) = 0\big\}. \tag{20}$$

Equivalently, it is the smallest $c$ such that $X \leq c$ holds almost surely (i.e., up to a $\mathbb{P}$-null event). When $X = f(\epsilon)$ with $\epsilon \sim \rho$, we write

$$\text{ess sup}_{\epsilon \sim \rho} f(\epsilon) := \inf\big\{c \in \mathbb{R} : \rho(\{\epsilon : f(\epsilon) > c\}) = 0\big\}.$$

For a distribution $\nu \in \mathscr{P}(\mathbb{R})$, we define its essential supremum by

$$\text{ess sup}(\nu) := \inf\big\{c \in \mathbb{R} : \nu(\{x \in \mathbb{R} : x > c\}) = 0\big\}. \tag{21}$$

If $Z \sim \nu$, then $\text{ess sup}(\nu) = \text{ess sup } Z$.

**A.2. Measure-Theoretic Definition of $\mathcal{T}_n^\pi$**

**Noise space used by the policy and the value.** Fix a noise dimension $d \in \mathbb{N}$ and define the base noise space $(\mathbb{R}^d, \mathcal{B}(\mathbb{R}^d), \rho)$, where $\rho = \mathcal{N}(0, I_d)$ is the standard Gaussian measure. When we write $\epsilon \sim \rho$, we mean that $\epsilon$ is a random vector with distribution $\rho$. Concretely, taking $(\Omega, \mathcal{F}, \mathbb{P}) = (\mathbb{R}^d, \mathcal{B}(\mathbb{R}^d), \rho)$ and $\epsilon(\omega) = \omega$ (the identity map) yields $\epsilon \sim \mathcal{N}(0, I_d)$ by construction. We will use two independent noise variables:

$$\epsilon_p \sim \rho \quad \text{(policy noise)}, \qquad \epsilon_v \sim \rho \quad \text{(value noise)}, \qquad \epsilon_p \perp \epsilon_v.$$

**Policy.** Our stochastic policy is a measurable mapping

$$\pi : \mathcal{S} \times \mathbb{R}^d \to \mathcal{A}, \qquad a = \pi(s, \epsilon_p).$$

The induced action distribution at state $s$ is

$$\pi(\cdot \mid s) = (\pi(s, \cdot))_{\#}\rho.$$

**Noise-conditioned Q-value.** A noise-conditioned critic is a measurable mapping

$$Q^\pi : \mathcal{S} \times \mathcal{A} \times \mathbb{R}^d \to \mathbb{R}, \qquad z = Q^\pi(s, a, \epsilon_v).$$

For fixed $(s, a)$, the quantity $Q^\pi(s, a, \epsilon_v)$ is a random variable induced by $\epsilon_v \sim \rho$, so the critic $Q^\pi(s, a, \cdot)$ induces a return distribution

$$\nu^\pi(s, a) := \left(Q^\pi(s, a, \cdot)\right)_{\#}\rho \in \mathscr{P}(\mathbb{R}).$$

Equivalently, for any Borel set $A \subset \mathbb{R}$, $\nu^\pi(s, a)(A) = \rho(\{\epsilon : Q^\pi(s, a, \epsilon) \in A\})$. Thus, repeatedly sampling $\epsilon \sim \rho$ and evaluating $Q^\pi(s, a, \epsilon)$ yields i.i.d. samples from $\nu^\pi(s, a)$.

**Affine Bellman shift.** For a reward $r \in \mathbb{R}$ and discount $\gamma \in [0, 1)$, define the measurable affine map

$$b_{r,\gamma} : \mathbb{R} \to \mathbb{R}, \qquad b_{r,\gamma}(z) = r + \gamma z.$$

**Standard distributional Bellman operator.** Given a collection of return distributions $\nu = \{\nu(s, a) \in \mathscr{P}(\mathbb{R})\}_{s,a}$, the standard distributional policy-evaluation operator (Eq.(7)) is

$$(\mathcal{T}^\pi \nu)(s, a) := \mathbb{E}_{s' \sim P(\cdot|s,a),\, \epsilon' \sim \rho}\left[(b_{r(s,a),\gamma})_{\#}\nu\left(s', \pi(s', \epsilon')\right)\right]. \tag{22}$$

It propagates the *entire* next-step return distribution through the Bellman backup.

**The proposed operator $\mathcal{T}_n^\pi$.** FAN replaces the next-step distribution by a Dirac mass at its upper-tail statistic $\operatorname{ess\,sup}(\nu(s', a'))$:

$$(\mathcal{T}_n^\pi \nu)(s, a) := \mathbb{E}_{s' \sim P(\cdot|s,a),\, \epsilon' \sim \rho}\left[(b_{r(s,a),\gamma})_{\#}\delta_{\operatorname{ess\,sup}\left(\nu(s', \pi(s', \epsilon'))\right)}\right]. \tag{23}$$

To organize, sample $s'$ from the environment and $\epsilon'$ from the noise, set $a' = \pi(s', \epsilon')$, compute the scalar $\operatorname{ess\,sup}(\nu(s', a'))$, place a Dirac mass at that scalar, and then apply the Bellman shift $z \mapsto r + \gamma z$.

**Connection to the sample-level equation (Eq. (9)).** Let $\epsilon' \sim \rho$ and define $a' = \pi(s', \epsilon')$. Let the transition dynamics be deterministic (i.e., $s'$ is fixed given $(s, a)$). The random scalar

$$Y_n(s, a, \epsilon') := r(s, a) + \gamma \operatorname{ess\,sup}\left(\nu^\pi(s', a')\right) = r(s, a) + \gamma \operatorname{ess\,sup}_{\epsilon \sim \rho} Q^\pi(s', a', \epsilon) \tag{24}$$

has distribution $\nu(Y_n(s, a, \epsilon')) = (\mathcal{T}_n^\pi \nu^\pi)(s, a)$ by Eq.(23). This is exactly the right-hand side of Eq.(9).

**A.3. Theoretical Motivations of $\mathcal{T}_n^\pi$**

We now provide theoretical motivations underlying $\mathcal{T}_n^\pi$.

**Motivation 1: Noise-conditioned critics represent distributions without tracking multiple statistics.** A noise-conditioned critic provides an implicit representation of the return distribution using a single function $Q^\pi(s, a, \epsilon)$, rather than explicitly maintaining multiple distributional statistics (e.g., a set of quantiles/expectiles). For each fixed $(s, a)$, the map $\epsilon \mapsto Q^\pi(s, a, \epsilon)$ defines a random variable with distribution

$$\nu^\pi(s, a) = \big(Q^\pi(s, a, \cdot)\big)_\# \rho \in \mathscr{P}(\mathbb{R}).$$

Thus, the critic can be viewed as a generative model for the return distribution, with $\epsilon$ serving as latent randomness.

If one plugs this representation into a standard distributional backup, a natural single-sample bootstrap target is

$$Y_{\text{std}} := r(s, a) + \gamma \, Q^\pi(s', a', \epsilon_v), \qquad a' = \pi(s', \epsilon_p), \ \ \epsilon_p \sim \rho, \ \ \epsilon_v \sim \rho,$$

which introduces an additional source of stochasticity through the critic-noise draw $\epsilon_v$ at the next-state evaluation. When only one (or a small number of) $\epsilon_v$ samples are used per transition, this *bootstrap noise* can dominate the variance of TD updates and slow finite-sample convergence. The operator $\mathcal{T}_n^\pi$ removes this particular source of variance by collapsing the next-step distribution using an upper-tail statistic.

**Motivation 2: Essential supremum removes critic-induced bootstrap noise (conditional variance reduction).** The essential supremum aggregates the next-step return distribution into a deterministic scalar:

$$\operatorname{ess\,sup}\big(\nu^\pi(s', a')\big) = \operatorname{ess\,sup}_{\epsilon \sim \rho} Q^\pi(s', a', \epsilon).$$

This scalar is then used in the Bellman target under $\mathcal{T}_n^\pi$.

To isolate the effect of critic-induced randomness, fix a transition $(s, a, r, s')$ and a next action $a' = \pi(s', \epsilon_p)$ for a given realization of $\epsilon_p$. Under the standard distributional backup, the target

$$Y_{\text{std}} = r(s, a) + \gamma \, Q^\pi(s', a', \epsilon_v), \qquad \epsilon_v \sim \rho,$$

retains randomness through $\epsilon_v$. Conditional on $(s', a')$, its variance is

$$\operatorname{Var}(Y_{\text{std}} \mid s', a') = \gamma^2 \operatorname{Var}(Q^\pi(s', a', \epsilon_v) \mid s', a').$$

In contrast, the $\mathcal{T}_n^\pi$ target

$$Y_n := r(s, a) + \gamma \, \operatorname{ess\,sup}_{\epsilon \sim \rho} Q^\pi(s', a', \epsilon)$$

is deterministic conditional on $(s', a')$, hence

$$\operatorname{Var}(Y_n \mid s', a') = 0.$$

Therefore, $\mathcal{T}_n^\pi$ *strictly reduces* the conditional variance attributable to bootstrap noise from critic sampling. Importantly, this statement does *not* claim that the overall target variance is zero, since randomness from environment transitions $s' \sim P(\cdot \mid s, a)$ and policy noise $a' = \pi(s', \epsilon_p)$ remains.

Variance control is central in stochastic approximation: finite-sample rates depend on the second moment of the update noise (Nemirovski et al., 2009; Moulines & Bach, 2011), and variance-reduced targets can improve sample efficiency in TD-style methods (Bhandari et al., 2018; Srikant & Ying, 2019).

**Motivation 3: Essential supremum yields a max-like, order-preserving utility for policy improvement.** In actor–critic methods, the critic is used to rank actions and guide policy improvement. In classical (risk-neutral) control, greedy policy improvement selects actions according to

$$\pi_{\text{new}}(s) \in \arg\max_{a \in \mathcal{A}} Q^\pi(s, a), \tag{25}$$

which follows directly from standard policy iteration and value-based control methods (Sutton & Barto, 2018). More generally, optimal control in Markov decision processes is characterized by the Bellman optimality operator

$$(\mathcal{T}^\star Q)(s, a) = r(s, a) + \gamma \, \mathbb{E}_{s' \sim P(\cdot \mid s, a)} \Big[ \sup_{a'} Q(s', a') \Big], \tag{26}$$

where the supremum is required for general (possibly infinite or continuous) action spaces. This operator is monotone and order-preserving with respect to $Q$ under standard assumptions, a fundamental property underpinning dynamic programming and reinforcement learning theory (Puterman, 1994; Bertsekas & Tsitsiklis, 1996).

With a distributional critic, each action $a$ induces a return distribution $\nu^\pi(s, a) = \big(Q^\pi(s, a, \cdot)\big)_{\#}\rho$. Consequently, policy improvement requires mapping the return distribution $\nu^\pi(s, a)$ to a scalar utility,

$$U^\pi(s, a) := \mathcal{F}\big(\nu^\pi(s, a)\big), \qquad \pi_{\text{new}}(s) \in \arg\max_{a \in \mathcal{A}} U^\pi(s, a).$$

We choose the essential supremum functional

$$U^\pi_{\max}(s, a) := \operatorname{ess\,sup}\big(\nu^\pi(s, a)\big) = \operatorname{ess\,sup}_{\epsilon \sim \rho} Q^\pi(s, a, \epsilon), \tag{27}$$

as a principled extension of the classical greedy policy improvement rule. In standard (risk-neutral) control, greedy improvement relies on maximizing scalar action-values, and the Bellman optimality operator itself is defined through a supremum over actions (Eq. (26)). The essential supremum preserves this maximization structure when action-values are represented as distributions rather than scalars: it reduces each return distribution to a single score that is compatible with max-based, order-preserving policy improvement. In this sense, $\operatorname{ess\,sup}$ serves as the natural distributional analogue of the classical $\max$ operator, ensuring conceptual continuity between scalar and distributional critics.

---

**Summary of Theoretical Motivations in $\mathcal{T}^\pi_n$ Design**

- **Implicit distribution representation.** $Q^\pi(s, a, \epsilon)$ represents return distributions without explicitly tracking multiple distributional statistics (e.g., multiple quantiles/expectiles).

- **Variance reduction in Bellman targets.** $\mathcal{T}^\pi_n$ removes critic-induced bootstrap noise, yielding lower-variance Bellman targets in temporal-difference updates.

- **Compatibility with greedy policy improvement.** $\operatorname{ess\,sup}$ preserves max-like policy improvement.

Together, these properties motivate $\mathcal{T}^\pi_n$ as a variance-reduced, distribution-aware Bellman operator that remains faithful to the core principles of greedy policy optimization.

# B. Theoretical Guarantees

## B.1. Convergence of the proposed operator $\mathcal{T}_n^\pi$ (Theorem 4.1)

**Definition B.1** (Supremum metric)**.** Let $\mathcal{Q}$ denote the space of bounded, measurable functions $Q : \mathcal{S} \times \mathcal{A} \times \mathbb{R}^d \to \mathbb{R}$. We define the metric $d_\infty$ on $\mathcal{Q}$ by

$$d_\infty(Q_1, Q_2) := \sup_{s \in \mathcal{S},\, a \in \mathcal{A},\, \epsilon \in \mathbb{R}^d} |Q_1(s, a, \epsilon) - Q_2(s, a, \epsilon)|.$$

> **Theorem 4.1 (Convergence of $\mathcal{T}_n^\pi$).** The proposed operator $\mathcal{T}_n^\pi$ is a $\gamma$-contraction on $(\mathcal{Q}, d_\infty)$ (Definition B.1), and therefore, iterating $\mathcal{T}_n^\pi$ from any $Q \in \mathcal{Q}$ converges to a unique fixed point $Q^\pi$.

*Proof.* Recall the definition of the proposed operator:

$$(\mathcal{T}_n^\pi Q)(s, a, \epsilon') = r(s, a) + \gamma\, \mathbb{E}_{s' \sim P(\cdot|s,a)} \left[ \operatorname*{ess\,sup}_\epsilon Q(s',\, \pi(s', \epsilon'),\, \epsilon) \right].$$

Since rewards are bounded and $Q \in \mathcal{Q}$ is bounded, $\mathcal{T}_n^\pi Q$ is also bounded, and hence $\mathcal{T}_n^\pi : \mathcal{Q} \to \mathcal{Q}$.

Let $Q_1, Q_2 \in \mathcal{Q}$. Using Definition B.1, we have

$$d_\infty(\mathcal{T}_n^\pi Q_1, \mathcal{T}_n^\pi Q_2) = \sup_{s,a,\epsilon'} |(\mathcal{T}_n^\pi Q_1)(s, a, \epsilon') - (\mathcal{T}_n^\pi Q_2)(s, a, \epsilon')|$$

$$\textit{(Bounded rewards cancel)} \quad = \sup_{s,a,\epsilon'} \left| \gamma \mathbb{E}_{s'} \left[ \operatorname*{ess\,sup}_\epsilon Q_1(s', \pi(s', \epsilon'), \epsilon) - \operatorname*{ess\,sup}_\epsilon Q_2(s', \pi(s', \epsilon'), \epsilon) \right] \right|$$

$$\textit{(Jensen's Inequality)} \quad \leq \gamma \sup_{s,a,\epsilon'} \mathbb{E}_{s'} \left[ \left| \operatorname*{ess\,sup}_\epsilon Q_1(s', \pi(s', \epsilon'), \epsilon) - \operatorname*{ess\,sup}_\epsilon Q_2(s', \pi(s', \epsilon'), \epsilon) \right| \right]$$

$$(|\sup f - \sup g| \leq \sup |f - g|) \quad \leq \gamma \sup_{s,a,\epsilon'} \mathbb{E}_{s'} \left[ \operatorname*{ess\,sup}_\epsilon |Q_1(s', \pi(s', \epsilon'), \epsilon) - Q_2(s', \pi(s', \epsilon'), \epsilon)| \right]$$

$$\textit{(Bound by max error over entire domain)} \quad \leq \gamma \sup_{s,a,\epsilon'} \mathbb{E}_{s'} \left[ \sup_{\hat{s},\hat{a},\hat{\epsilon}} |Q_1(\hat{s}, \hat{a}, \hat{\epsilon}) - Q_2(\hat{s}, \hat{a}, \hat{\epsilon})| \right]$$

Since the inner expression of the last equation is uniformly bounded by $d_\infty(Q_1, Q_2)$ over the entire domain, and the expectation of a constant is the constant itself, we conclude

$$d_\infty(\mathcal{T}_n^\pi Q_1, \mathcal{T}_n^\pi Q_2) \leq \gamma\, d_\infty(Q_1, Q_2).$$

Thus, $\mathcal{T}_n^\pi$ is a $\gamma$-contraction on $(\mathcal{Q}, d_\infty)$. Since $\mathcal{Q}$ equipped with the supremum norm is a complete metric space, Banach's Fixed Point Theorem (Banach, 1922) guarantees the existence of a unique fixed point $Q^\pi$, and that iterating $\mathcal{T}_n^\pi$ from any initial $Q \in \mathcal{Q}$ converges to $Q^\pi$. $\square$

## B.2. The Upper Expectile converges to the Essential Supremum (Theorem 4.2)

**Lemma B.2** (**Basic Properties of Expectiles**). *Let $X$ be a real-valued random variable with $\mathbb{E}[X^2] < \infty$ and let $\kappa \in (0, 1)$. Define the asymmetric least-squares loss*

$$\mathcal{L}_2^\kappa(u) := |\kappa - \mathbf{1}(u < 0)|\, u^2 = \kappa\, u_+^2 + (1 - \kappa)\, u_-^2, \quad u_+ := \max\{u, 0\}, \ \ u_- := \max\{-u, 0\},$$

*and the $\kappa$-expectile*

$$e_\kappa(X) := \arg\min_{q \in \mathbb{R}} \mathbb{E}\big[\mathcal{L}_2^\kappa(X - q)\big].$$

*Then:*

  *(i) (**Mean as a special case**) $e_{1/2}(X) = \mathbb{E}[X]$.*

  *(ii) (**Non-decreasing over $\kappa$**) The map $\kappa \mapsto e_\kappa(X)$ is non-decreasing on $(0, 1)$.*

  *(iii) (**Range bound**) If $X$ is essentially bounded with $m \le \operatorname{ess\,inf} X$ and $\operatorname{ess\,sup} X \le M$, then $e_\kappa(X) \in [m, M]$.*

*Proof.* **Existence and uniqueness.** Since $q \mapsto \mathcal{L}_2^\kappa(X - q)$ is convex for each $X$ and strictly convex in $q$ on any event with positive probability (because the quadratic has strictly positive curvature on both sides), the objective $q \mapsto \mathbb{E}[\mathcal{L}_2^\kappa(X - q)]$ is strictly convex on $\mathbb{R}$. Hence, the minimizer $e_\kappa(X)$ exists and is unique.

**(i) Mean at $\kappa = \frac{1}{2}$.** For $\kappa = \frac{1}{2}$,

$$\mathcal{L}_2^{1/2}(X - q) = \tfrac{1}{2}(X - q)^2,$$

so the unique minimizer is $q = \mathbb{E}[X]$.

**A useful characterization (first-order condition).** For $\kappa \in (0, 1)$, differentiating the objective w.r.t. $q$ (valid since $\mathbb{E}[X^2] < \infty$) yields the necessary and sufficient optimality condition at $q = e_\kappa(X)$:

$$\kappa\, \mathbb{E}\big[(X - q)_+\big] = (1 - \kappa)\, \mathbb{E}\big[(q - X)_+\big], \qquad q = e_\kappa(X). \tag{28}$$

We will use Eq.(28) for (ii) and (iii).

**(ii) Non-decreasing over $\kappa$.** Fix $\kappa_1 < \kappa_2$ and denote $q_i := e_{\kappa_i}(X)$. Suppose for contradiction that $q_2 < q_1$. Note that

$$A(q) := \mathbb{E}[(X - q)_+] \quad \text{is non-increasing in } q, \text{ and} \qquad B(q) := \mathbb{E}[(q - X)_+] \quad \text{is non-decreasing in } q.$$

Hence $q_2 < q_1$ implies $A(q_2) \ge A(q_1)$ and $B(q_2) \le B(q_1)$. Using the first-order condition Eq.(28) for each $\kappa_i$ gives

$$\frac{A(q_i)}{B(q_i)} = \frac{1 - \kappa_i}{\kappa_i}, \qquad i \in \{1, 2\}.$$

But since $q_2 < q_1$, we have

$$\frac{A(q_2)}{B(q_2)} \ge \frac{A(q_1)}{B(q_1)}.$$

On the other hand, $\kappa \mapsto \frac{1-\kappa}{\kappa}$ is strictly decreasing on $(0, 1)$, so

$$\frac{A(q_2)}{B(q_2)} = \frac{1 - \kappa_2}{\kappa_2} < \frac{1 - \kappa_1}{\kappa_1} = \frac{A(q_1)}{B(q_1)},$$

a contradiction. Therefore $q_2 \ge q_1$, proving that $\kappa \mapsto e_\kappa(X)$ is non-decreasing.

**(iii) Range bound.** Assume $\operatorname{ess\,sup} X \le M$. For any $q > M$, we have $X - q < 0$ almost surely, so $(X - q)_+ = 0$ and $(q - X)_+ = q - X > 0$ a.s., implying the left side of Eq.(28) is 0 while the right side is strictly positive. Hence Eq.(28) cannot hold for $q > M$, so $e_\kappa(X) \le M$. Similarly, if $\operatorname{ess\,inf} X \ge m$, then for any $q < m$ we have $(q - X)_+ = 0$ and $(X - q)_+ = X - q > 0$ a.s., so Eq.(28) cannot hold meaning that $e_\kappa(X) \ge m$. Therefore, $e_\kappa(X) \in [m, M]$. $\qquad\square$

**Theorem 4.2 (Upper Expectile Converges to the Essential Supremum)** Let $s \in \mathcal{S}$, $a \in \mathcal{A}$, $\epsilon \sim \mathcal{N}(0, I_d)$, and $Q \in \mathcal{Q}$. For any $\kappa \in [\frac{1}{2}, 1)$, $Z_\kappa := \arg\min_{q \in \mathbb{R}} \mathbb{E}_\epsilon\left[\mathcal{L}_2^\kappa(Q(s, a, \epsilon) - q)\right]$ is bounded by:

$$Z_{1/2} \leq Z_\kappa \leq \lim_{\kappa \to 1^-} Z_\kappa = \operatorname{ess\,sup}_\epsilon Q(s, a, \epsilon). \tag{29}$$

*Proof.* We proceed in two steps: (i) establish the sandwich bounds and existence of the limit, and (ii) identify the limit with the essential supremum.

**Step 1: Sandwich bounds and existence of the limit.** Let

$$X := Q(s, a, \epsilon), \qquad \epsilon \sim \mathcal{N}(0, I_d),$$

and denote $Z_\kappa := e_\kappa(X)$. By Lemma B.2 **(ii)**, the map $\kappa \mapsto Z_\kappa$ is non-decreasing on $(0, 1)$. In particular, for all $\kappa \in [\frac{1}{2}, 1)$,

$$Z_{1/2} \leq Z_\kappa \leq \sup_{\kappa < 1} Z_\kappa.$$

Moreover, since $X$ is essentially bounded and $M := \operatorname{ess\,sup}_\epsilon X < \infty$, Lemma B.2 **(iii)** implies $Z_\kappa \leq M$ for all $\kappa \in (0, 1)$. Therefore, the monotone limit

$$Z^* := \lim_{\kappa \to 1^-} Z_\kappa = \sup_{\kappa < 1} Z_\kappa$$

exists and satisfies $Z^* \leq M$. This proves the first two inequalities in Eq.(29).

**Step 2: Identification of the limit.** By the first-order optimality condition (Lemma B.2, Eq. (28)), $Z_\kappa$ satisfies

$$\kappa \, \mathbb{E}[(X - Z_\kappa)_+] = (1 - \kappa) \, \mathbb{E}[(Z_\kappa - X)_+]. \tag{30}$$

Since $X \in [m, M]$ and $Z_\kappa \in [m, M]$ almost surely, we have $(Z_\kappa - X)_+ \leq M - m$, and thus

$$\mathbb{E}[(Z_\kappa - X)_+] \leq M - m.$$

Substituting into (30) yields

$$0 \leq \mathbb{E}[(X - Z_\kappa)_+] = \frac{1 - \kappa}{\kappa} \, \mathbb{E}[(Z_\kappa - X)_+] \leq \frac{1 - \kappa}{\kappa} \, (M - m) \xrightarrow[\kappa \to 1^-]{} 0. \tag{31}$$

Now suppose, for contradiction, that $Z^* < M$. Then there exists $\delta > 0$ such that $Z^* \leq M - \delta$. Since $Z^*$ is the non-decreasing limit of $Z_\kappa$, there exists $\kappa_0$ such that $Z_\kappa \leq M - \delta$ for all $\kappa \geq \kappa_0$. Hence, for all such $\kappa$,

$$(X - Z_\kappa)_+ \geq (X - (M - \delta))_+, \quad \text{and therefore} \quad \mathbb{E}[(X - Z_\kappa)_+] \geq \mathbb{E}[(X - (M - \delta))_+].$$

By the definition of the essential supremum, $\mathbb{P}(X > M - \delta) > 0$, which implies

$$C_\delta := \mathbb{E}[(X - (M - \delta))_+] > 0.$$

Thus, for all $\kappa \geq \kappa_0$,

$$\mathbb{E}[(X - Z_\kappa)_+] \geq C_\delta > 0,$$

which contradicts Eq.(31). Therefore, the assumption $Z^* < M$ is false, and we conclude

$$\lim_{\kappa \to 1^-} Z_\kappa = M = \operatorname{ess\,sup}_\epsilon Q(s, a, \epsilon).$$

Combining with Step 1 completes the proof of Eq.(29). □

## B.3. Validity of Flow Anchoring as Behavior Regularization (Theorem 4.3)

> **Lemma B.3** (Derivative of the Norm Bound). *Let $e : [0, 1] \to \mathbb{R}^d$ be an absolutely continuous function and let $g(t) := \|e(t)\|_2$. Then $g$ is absolutely continuous and its derivative satisfies:*
>
> $$g'(t) \leq \left\| \frac{d}{dt} e(t) \right\|_2 \quad \text{for almost every } t \in [0, 1]. \tag{32}$$

*Proof.* Since $e(t)$ is absolutely continuous, it is differentiable almost everywhere. At any point $t$ where $e(t)$ is differentiable and $e(t) \neq 0$, we apply the chain rule to the squared norm $g(t)^2 = \langle e(t), e(t) \rangle$:

$$\frac{d}{dt}(g(t)^2) = \frac{d}{dt}\langle e(t), e(t) \rangle = 2 \left\langle e(t), \frac{d}{dt} e(t) \right\rangle. \tag{33}$$

On the other hand, applying the chain rule to the scalar function $g(t)^2$ directly yields:

$$\frac{d}{dt}(g(t)^2) = 2g(t)g'(t) = 2\|e(t)\|_2 \, g'(t). \tag{34}$$

Equating the two expressions gives:

$$\|e(t)\|_2 \, g'(t) = \left\langle e(t), \frac{d}{dt} e(t) \right\rangle. \tag{35}$$

Using the Cauchy-Schwarz inequality, $\langle a, b \rangle \leq \|a\|_2 \|b\|_2$, we obtain:

$$\|e(t)\|_2 \, g'(t) \leq \|e(t)\|_2 \left\| \frac{d}{dt} e(t) \right\|_2. \tag{36}$$

Since we assumed $\|e(t)\|_2 > 0$, we can divide both sides by $\|e(t)\|_2$ to get:

$$g'(t) \leq \left\| \frac{d}{dt} e(t) \right\|_2. \tag{37}$$

For the case where $e(t) = 0$, the inequality holds trivially if interpreted in the sense of generalized derivatives or by limits, as the minimum of the norm implies a derivative of zero or undefined (but bounded by the directional derivative). Since $e$ is absolutely continuous, this relation holds for almost every $t \in [0, 1]$. $\square$

> **Lemma B.4** (Differential Grönwall inequality). *Let $g : [0, 1] \to \mathbb{R}_{\geq 0}$ be absolutely continuous and suppose*
>
> $$\frac{d}{dt}g(t) \leq L g(t) + b(t) \quad \text{for almost every } t \in [0, 1], \tag{38}$$
>
> *where $L \geq 0$ is a constant and $b(t) \geq 0$ is integrable. Then for all $t \in [0, 1]$, $g(t) \leq e^{Lt} g(0) + \int_0^t e^{L(t-u)} b(u) \, du$. In particular, if $g(0) = 0$, then*
>
> $$g(t) \leq \int_0^t e^{L(t-u)} b(u) \, du \leq e^{Lt} \int_0^t b(u) \, du. \tag{39}$$

*Proof.* Define $h(t) := e^{-Lt} g(t)$. Since $g$ is absolutely continuous, so is $h$, and for almost every $t$,

$$\frac{d}{dt} h(t) = e^{-Lt} \left( \frac{d}{dt} g(t) - Lg(t) \right) \leq e^{-Lt} b(t).$$

Integrating from 0 to $t$ yields $h(t) - h(0) \leq \int_0^t e^{-Lu} b(u) \, du$, and therefore, $g(t) \leq e^{Lt} g(0) + e^{Lt} \int_0^t e^{-Lu} b(u) \, du$. $\square$

**Definition B.5** (Induced distributions $\mu_\omega, \mu_\theta$ by the one-step policy $\pi_\omega$ and the behavior flow policy $v_\theta$). For $s \in \mathcal{S}$, the one-step policy $\pi_\omega$ induces the distribution $\mu_\theta(\cdot|s)$:

$$\mu_\omega(\cdot|s) := (\pi_\omega(s, \cdot))_\# \mathcal{N}(0, I_d),$$

modeling the action distribution of the one-step policy. Likewise, the behavior flow policy $v_\theta$ defines the ODE:

$$\frac{dx_t}{dt} = v_\theta(s, t, x_t), \quad t \in [0, 1],$$

where $x_t := x_\theta(s, t, z)$ is the state of the flow at time $t$, following $v_\theta(s, t, x_t)$ with $t \sim \text{Unif}([0, 1])$, starting from $x_0 = x_\theta(s, 0, z) = z \sim \mathcal{N}(0, I_d)$. The flow map $\Phi_\theta(s, z) := x_1 = x_\theta(s, 1, z)$ induces the distribution:

$$\mu_\theta(\cdot|s) := (\Phi_\theta(s, \cdot))_\# \mathcal{N}(0, I_d),$$

which models the offline dataset behavior distribution.

**Assumption B.6** (Lipschitz behavior vector field). $L \geq 0$ exists for all $s \in \mathcal{S}, t \in [0, 1]$, and $x, y \in \mathcal{A}$, satisfying:

$$\|v_\theta(s, t, x) - v_\theta(s, t, y)\|_2 \leq L\|x - y\|_2.$$

---

**Lemma B.7** (Endpoint mismatch is controlled by the flow residual). *Assume $v_\theta$ satisfies Assumption B.6. Let $s \in \mathcal{S}$, $t \in [0, 1]$, $z \in \mathbb{R}^d$, and $x_\theta(s, t, z)$ solve $\frac{d}{dt}x_\theta(s, t, z) = v_\theta(s, t, x_\theta(s, t, z))$ with $x_\theta(s, 0, z) = z$. Let $y(s, t, z)$ be any absolutely continuous path with $y(s, 0, z) = z$. Then, the endpoint deviation satisfies:*

$$\|y(s, 1, z) - x_\theta(s, 1, z)\|_2^2 \leq e^{2L} \int_0^1 \|\frac{d}{dt}y(s, t, z) - v_\theta(s, t, y(s, t, z))\|_2^2 \, dt. \tag{40}$$

---

*Proof.* Define the error $e(s, t, z) := y(s, t, z) - x_\theta(s, t, z)$, so $e(s, 0, z) = 0$. Using $\frac{d}{dt}x_\theta(s, t, z) = v_\theta(s, t, x_\theta(s, t, z))$:

$$\frac{d}{dt}e(s, t, z) = \frac{d}{dt}y(s, t, z) - \frac{d}{dt}x_\theta(s, t, z) = \frac{d}{dt}y(s, t, z) - v_\theta(s, t, x_\theta(s, t, z))$$

$$= \underbrace{\left(\frac{d}{dt}y(s, t, z) - v_\theta(s, t, y(s, t, z))\right)}_{r(s,t,z)} + \underbrace{\left(v_\theta(s, t, y(s, t, z)) - v_\theta(s, t, x_\theta(s, t, z))\right)}_{\Delta(s,t,z)}.$$

Taking norms and applying Triangular inequality and Lipschitzness (Assumption B.6) gives:

$$\|\frac{d}{dt}e(s, t, z)\|_2 \leq \|r(s, t, z)\|_2 + \|\Delta(s, t, z)\|_2 \leq \|r(s, t, z)\|_2 + L\|e(s, t, z)\|_2. \tag{41}$$

Let $g(s, t, z) := \|e(s, t, z)\|_2$. Since $e$ is absolutely continuous, $g$ is absolutely continuous and satisfies $\frac{d}{dt}g(s, t, z) \leq \|\frac{d}{dt}e(s, t, z)\|_2$ for almost every $t$. Combining this inequality with Eq.(41) yields

$$\frac{d}{dt}g(s, t, z) \leq \|\frac{d}{dt}e(s, t, z)\|_2 \leq \|r(s, t, z)\|_2 + L\|e(s, t, z)\|_2 = \|r(s, t, z)\|_2 + Lg(s, t, z) \quad \text{for almost every } t \in [0, 1],$$

With $b(s, t, z) = \|r(s, t, z)\|_2$ and $g(s, 0, z) = \|e(s, 0, z)\|_2 = 0$, we can apply Lemma B.4 by satisfying Eq.(38). Therefore, by Eq.(39),

$$\|e(s, 1, z)\|_2 = g(s, 1, z) \leq e^L \int_0^1 \|r(s, t, z)\|_2 \, dt.$$

Finally, Cauchy–Schwarz yields

$$\|e(s, 1, z)\|_2^2 \leq e^{2L} \left(\int_0^1 \|r(s, t, z)\|_2 \, dt\right)^2 \leq e^{2L} \int_0^1 \|r(s, t, z)\|_2^2 \, dt.$$

Since $e(s, 1, z) = y(s, 1, z) - x_\theta(s, 1, z)$ and $r(s, t, z) = \frac{d}{dt}y(s, t, z) - v_\theta(t, y(s, t, z), s)$, this proves Eq.(40). $\square$

> **Theorem 4.3 (Flow Anchoring is a Valid Behavior Regularization)** Let $\mu_\omega(\cdot|s)$ and $\mu_\theta(\cdot|s)$ be the probability distributions induced by the policy $\pi_\omega$ and the behavior flow $v_\theta$ respectively (Definition B.5). If $v_\theta$ satisfies Lipschitzness (Assumption B.6), the following holds for all $s \in \mathcal{S}$:
>
> $$\mathbb{E}_{s\sim\mathcal{D}}\Big[W_2^2(\mu_\omega(\cdot|s),\, \mu_\theta(\cdot|s))\Big] \;\leq\; e^{2L}\,\mathcal{L}_B(\omega), \tag{42}$$
>
> where $W_2$ is the Wasserstein-2 distance and $L$ is the Lipschitz constant.

*Proof.* Since $W_2$ is the infimum over all couplings, the following inequality holds with $\Phi_\theta$ following Definition B.5:

$$W_2^2(\mu_\omega(\cdot|s),\, \mu_\theta(\cdot|s)) := \inf_{\gamma\in\Gamma(\mu_\omega,\mu_\theta)} \mathbb{E}_{(A,B)\sim\gamma}\big[\|A-B\|_2^2\big] \leq \mathbb{E}_z\big[\|\pi_\omega(s,z) - \Phi_\theta(s,z)\|_2^2\big],$$

where $\Gamma(\cdot,\cdot)$ is the set of couplings with the input marginals. Also, following Lemma B.7 with $y(s,t,z) = (1-t)z + t\pi_\omega(s,z)$ leads to:

$$\|y(s,1,z) - x_\theta(s,1,z)\|_2^2 = \|\pi_\omega(s,z) - \Phi_\theta(s,z)\|_2^2 \leq e^{2L}\int_0^1 \|(\pi_\omega(s,z) - z) - v_\theta(s,t,(1-t)z + t\pi_\omega(s,z))\|_2^2\, dt. \tag{43}$$

Therefore,

$$\mathbb{E}_{s\sim\mathcal{D}}\Big[W_2^2(\mu_\omega(\cdot|s),\, \mu_\theta(\cdot|s))\Big] \;\leq\; \mathbb{E}_{\substack{s\sim\mathcal{D},\\ z\sim\mathcal{N}(0,I_d)}} \big[\|\pi_\omega(s,z) - \Phi_\theta(s,z)\|_2^2\big] \tag{44}$$

$$\leq\; e^{2L}\,\mathbb{E}_{\substack{s\sim\mathcal{D},\\ z\sim\mathcal{N}(0,I_d),\\ t\sim\mathrm{Unif}([0,1])}} \big[\|(\pi_\omega(s,z) - z) - v_\theta(s,t,(1-t)z + t\pi_\omega(s,z))\|_2^2\big] = e^{2L}\mathcal{L}_B(\omega) \tag{45}$$

Given $s \in \mathcal{S}$, the equality holds when $\mu_\omega(\cdot|s) = \mu_\theta(\cdot|s)$ and all flow trajectories of the vector field $v_\theta$ are straight. This is because it is the case when $W_2^2(\mu_\omega(\cdot|s), \mu_\theta(\cdot|s)) = 0$ and $\pi_\omega(s,z) - z = v_\theta(s,t,(1-t)z + t\pi_\omega(s,z))$ satisfies for all $z \sim \mathcal{N}(0,I_d)$ and $t \sim \mathrm{Unif}([0,1])$. $\qquad\square$

# C. Experimental Details

We implement FAN using JAX (Bradbury et al., 2018), building upon the code implementations of FQL (Park et al., 2025c) and Value Flows (Dong et al., 2026). We adopt these frameworks for two reasons: first, FQL provides the fastest training and inference speeds among flow policy-based methods; and second, Value Flows achieves the highest performance among distributional methods that utilize flow policies.

## C.1. Benchmarks

**D4RL.** D4RL (Fu et al., 2020) is a well-established standard for benchmarking offline RL algorithms. Specifically, we measure normalized returns to compare performance on the relatively harder tasks in this benchmark. Therefore, we include 4 `antmaze` tasks involving 8-DoF locomotion, and 12 `adroit` tasks involving dexterous manipulation (i.e., $\geq$ 24-DoF).

1. Antmaze Datasets

   - `antmaze-medium-play-v2`
   - `antmaze-medium-diverse-v2`
   - `antmaze-large-play-v2`
   - `antmaze-large-diverse-v2`

2. Adroit Datasets

   - `pen-human-v1`
   - `pen-cloned-v1`
   - `pen-expert-v1`
   - `door-human-v1`
   - `door-cloned-v1`
   - `door-expert-v1`
   - `hammer-human-v1`
   - `hammer-cloned-v1`
   - `hammer-expert-v1`
   - `relocate-human-v1`
   - `relocate-cloned-v1`
   - `relocate-expert-v1`

The Antmaze tasks require controlling a quadrupedal agent to reach a goal in a given maze. The Adroit tasks require learning complex skills such as spinning a pen, opening a door, relocating a ball, and using a hammer to hit a button.

**OGBench.** OGBench (Park et al., 2025a) was originally designed for offline goal-conditioned RL. However, this benchmark also provides single-task variants to benchmark standard reward-maximizing offline RL approaches. Therefore, we use 27 state-based and 4 pixel-based single-tasks in OGBench, particularly focusing on environments where prior offline RL methods struggle to achieve 100% success rates. To label transition rewards in the dataset, these single-tasks apply semi-sparse reward functions, where the function is defined as the negative of the number of remaining subtasks at a given state. Locomotion tasks involve a single subtask, and the rewards are always $-1$ or $0$. Manipulation tasks normally include multiple subtasks, so the rewards are bounded by $-n_{\text{subtask}}$ (i.e., number of subtasks) and $0$. The following state-based and pixel-based datasets are used in our offline RL experiments:

1. 1M-sized State-based Datasets (5 tasks each)

   - `antsoccer-arena-navigate-v0`
   - `scene-play-v0`
   - `cube-double-play-v0`
   - `puzzle-3x3-play-v0`
   - `puzzle-4x4-play-v0`

2. 1M-sized Pixel-based Datasets (1 task each)

- `visual-antmaze-medium-navigate-v0`
- `visual-antmaze-teleport-navigate-v0`
- `visual-cube-double-play-v0`
- `visual-puzzle-4x4-play-v0`

We utilize these datasets to evaluate diverse RL capabilities, ranging from standard offline learning to visual control. For standard benchmarks, we employ five 1M-sized state-based tasks: `antsoccer-arena-navigate` for quadrupedal ball dribbling, `scene-play` for long-horizon object interaction, `cube-double-play` for pick-and-place manipulation, and `puzzle-3x3/4x4-play` for combinatorial generalization on "Lights Out" puzzles. To test representation learning under partial observability, we include 1M-sized pixel-based variants (`visual-antmaze`, `visual-cube`, `visual-puzzle`) that require control solely from $64 \times 64 \times 3$ images.

## C.2. Baseline Methods

We compare FAN to six prior approaches. The first three include computationally efficient non-distributional approaches that report near state-of-the-art performance, and the latter three include distributional approaches using flow policies. Note that IQN and CODAC were originally proposed using Gaussian policies, but we modified the algorithms to use flow policies, leading to better performance in our experience. We fix learning rates ($3\mathrm{e}{-}4$) and target update rates ($5\mathrm{e}{-}3$), and use 8 seeds for state-based training and 4 seeds for pixel-based task training. For pixel-based tasks, we additionally use the IMPALA encoder (Espeholt et al., 2018) for state representations.

**ReBRAC.** ReBRAC (Tarasov et al., 2023a) is an offline actor-critic algorithm building on TD3+BC (Fujimoto & Gu, 2021) that incorporates architectural enhancements such as layer normalization and critic decoupling. The algorithm relies on two primary hyperparameters: $\alpha_1$, which controls the strength of the actor behavior cloning (BC) regularization, and $\alpha_2$, which governs the critic BC regularization. Consistent with the baselines in FQL and Value Flows, we directly report the results from Park et al. (2025c) and Dong et al. (2026). We report the best performance between using flow-based behavior regularization with 10 flow steps (i.e., FBRAC in Park et al. (2025c)) and the standard one in Tarasov et al. (2023a).

**IDQL.** Implicit Diffusion Q-Learning (IDQL) (Hansen-Estruch et al., 2023) decouples value learning from policy extraction by combining IQL (Kostrikov et al., 2022) with a diffusion-based behavior model. During inference, the agent samples $N$ action candidates and selects the one maximizing the learned Q-value. We also include IFQL (Park et al., 2025c) in this category, a variant that replaces the diffusion component with a flow matching policy. Consistent with other baselines, we report results directly from Park et al. (2025c) and Dong et al. (2026), selecting the best performance between IDQL and IFQL for each task. We use 10 steps for diffusion or flow policy sampling.

**FQL.** Flow Q-Learning (FQL) (Park et al., 2025c) utilizes a one-step flow policy to maximize Q-value estimates learned via standard TD error. FQL incorporates a behavioral regularization term with coefficient $\alpha$ towards a behavior-cloning flow policy (Eq.6). We also directly report the results from Park et al. (2025c) and Dong et al. (2026) that use 10 flow steps.

**IQN.** Implicit Quantile Networks (IQN) (Dabney et al., 2018a) is a distributional RL method that approximates the return distribution by predicting quantile values at randomly sampled quantile fractions. Following Dong et al. (2026), we apply 10 flow step rejection sampling to the flow policy for inference, using 16 noise and 16 quantile samples. We perform a hyperparameter sweep for the temperature $\kappa$ in the quantile regression loss over the values $\{0.7, 0.8, 0.9, 0.95\}$.

**CODAC.** Conservative Offline Distributional Actor Critic (CODAC) (Ma et al., 2021) augments the distributional critic of IQN with conservative constraints. Following Dong et al. (2026), we utilize a one-step flow policy regularized through actions sampled with 10 flow steps, which follows a DDPG-style policy extraction. We fix the conservative penalty coefficient to 0.1 and tune the remaining hyperparameters by sweeping the quantile regression loss temperature $\kappa \in \{0.7, 0.8, 0.9, 0.95\}$ and the BC coefficient $\alpha_1 \in \{100, 300, 1000, 3000, 10000, 30000\}$.

**Value Flows.** Value Flows (Dong et al., 2026) is a distributional RL algorithm that leverages flow matching to estimate the full distribution of future returns. By formulating a distributional flow matching objective, it learns a return vector field that satisfies the distributional Bellman equation. For offline policy extraction, it employs 10 flow step rejection sampling with a behavioral cloning flow policy to select actions that maximize expected returns. The key difference with FAN is that Value Flows uses 1-dimensional Gaussian noise to match with the dimensions of rewards. We sweep the regularization coefficient $\lambda \in \{0.3, 1, 3, 10\}$ and the confidence weight temperature $\tau \in \{0.01, 0.03, 0.1, 0.3, 1\}$ for results not in Dong et al. (2026).

**FAN.** Following prior work, we standardize the architecture to [512, 512, 512, 512]-sized MLPs for all networks (e.g., one-step policy, value, behavioral flow policy). Also, we use a fixed expectile $\kappa = 0.9$ across all tasks, and sweep only $\alpha_1$ and $\alpha_2$. For OGBench tasks, we sweep $\alpha_1 \in \{10, 30, 100, 300\}$ and $\alpha_2 \in \{0, 0.1, 0.3, 1, 3\}$. For D4RL `antmaze` tasks, we sweep $\alpha_1 \in \{1, 3, 10\}$ and $\alpha_2 \in \{0, 0.01, 0.03, 0.1, 0.3, 1\}$. For D4RL `adroit` tasks, we sweep $\alpha_1 \in \{1000, 3000, 10000, 30000\}$ and $\alpha_2 \in \{0, 0.1, 0.3, 1, 3, 10\}$. Such selection is intended to maintain $\alpha_1$ similar to $\alpha$ in Park et al. (2025c), and also similar to the hyperparameter choices in Dong et al. (2026).

**NBRAC, NFQL, FAQL.** For Ablation Study 1, we propose Noise-conditioned Behavior Regularized Actor Critic (NBRAC), a variant of ReBRAC using noise-conditioned critic. We maintain the behavior regularization of ReBRAC and substitute the standard Q-value update to $\mathcal{T}_n^\pi$. For Ablation Study 1, we also propose Noise-conditioned Flow Q-Learning (NFQL), a variant of FQL using noise-conditioned critic. We maintain the behavior regularization of FQL and substitute the standard Q-value update to $\mathcal{T}_n^\pi$. For Ablation Study 2, we propose Flow Anchored Q-Learning (FAQL), a variant of FQL using Flow Anchoring. We maintain the standard Q-value update and substitute the behavior regularization to Flow Anchoring.

## C.3. Hyperparameters

*Table 3.* **Hyperparameter Configurations for FAN** shared across experiments.

| Hyperparameter | Value |
|---|---|
| Learning rate | 0.0003 |
| Optimizer | Adam (Kingma & Ba, 2015) |
| Offline Gradient steps | 1000000 (default), 500000 (D4RL, pixel-based OGBench) |
| Offline-to-Online Gradient steps (Offline) | 1000000 |
| Offline-to-Online Gradient steps (Online) | 1000000 |
| Minibatch size | 256 |
| MLP dimensions | $[512, 512, 512, 512]$ |
| Nonlinearity | GELU (Hendrycks & Gimpel, 2016) |
| Target network smoothing coefficient | 0.005 |
| Expectile $\kappa$ | 0.9 |
| Discount factor $\gamma$ | 0.995 (default), 0.99 (D4RL) |
| Image augmentation probability | 0.5 |
| Flow time sampling distribution | Unif($[0, 1]$) |
| Number of Q ensembles | 2 |
| Number of Z ensembles | 2 |
| Target value aggregation | mean (default), min (D4RL, pixel-based OGBench) |
| Actor BC coefficient $\alpha_1$ | See Tables 4 to 6 |
| Critic BC coefficient $\alpha_2$ | See Tables 4 to 6 |

*Table 4.* **Detailed Hyperparameter Configurations for Offline Results.** We mostly take configurations from Park et al. (2025c) and Dong et al. (2026). For all baselines other than FAN, the discount factor $\gamma$ is 0.99 in default and 0.995 for `antsoccer`, `cube-double`, and `visual-cube-double` tasks. "-" indicates that the results are taken from prior work or excluded.

| | | ReBRAC | | IDQL | FQL | IQN | CODAC | | Value Flows | | FAN | |
|---|---|---|---|---|---|---|---|---|---|---|---|---|
| Benchmark | Task | $\alpha_1$ | $\alpha_2$ | $N$ | $\alpha$ | $\kappa$ | $\kappa$ | $\alpha$ | $\lambda$ | $\tau$ | $\alpha_1$ | $\alpha_2$ |
| D4RL (Antmaze) | Antmaze-medium-play-v2 | - | - | - | 10 | 0.8 | 0.95 | 3 | 3 | 1 | 3 | 0.01 |
| | Antmaze-medium-diverse-v2 | - | - | - | 10 | 0.8 | 0.9 | 3 | 10 | 1 | 3 | 0.01 |
| | Antmaze-large-play-v2 | - | - | - | 3 | 0.9 | 0.95 | 3 | 3 | 1 | 3 | 0.03 |
| | Antmaze-large-diverse-v2 | - | - | - | 3 | 0.9 | 0.9 | 3 | 1 | 1 | 3 | 0.03 |
| D4RL (Adroit) | Pen-human | - | - | 32 | 10000 | 0.8 | 0.8 | 10000 | 3 | 1 | 1000 | 1 |
| | Pen-cloned | - | - | 32 | 10000 | 0.8 | 0.8 | 10000 | 3 | 1 | 1000 | 0 |
| | Pen-expert | - | - | 32 | 3000 | 0.8 | 0.8 | 10000 | 3 | 0.01 | 1000 | 0 |
| | Door-human | - | - | 32 | 30000 | 0.9 | 0.9 | 10000 | 3 | 0.01 | 10000 | 10 |
| | Door-cloned | - | - | 32 | 30000 | 0.9 | 0.9 | 30000 | 10 | 0.3 | 3000 | 10 |
| | Door-expert | - | - | 32 | 30000 | 0.9 | 0.9 | 10000 | 10 | 0.3 | 3000 | 10 |
| | Hammer-human | - | - | 128 | 30000 | 0.7 | 0.8 | 30000 | 3 | 0.3 | 10000 | 1 |
| | Hammer-cloned | - | - | 32 | 10000 | 0.7 | 0.8 | 10000 | 3 | 0.3 | 10000 | 0.3 |
| | Hammer-expert | - | - | 32 | 30000 | 0.9 | 0.8 | 10000 | 10 | 1 | 10000 | 1 |
| | Relocate-human | - | - | 32 | 10000 | 0.9 | 0.9 | 30000 | 10 | 0.01 | 10000 | 10 |
| | Relocate-cloned | - | - | 32 | 30000 | 0.9 | 0.9 | 30000 | 3 | 0.01 | 10000 | 10 |
| | Relocate-expert | - | - | 32 | 30000 | 0.9 | 0.9 | 10000 | 3 | 0.1 | 30000 | 10 |
| OGBench (State-based) | Antsoccer-arena-navigate | 0.01 | 0.01 | 32 | 10 | 0.9 | 0.95 | 10 | 1 | 1 | 10 | 0.1 |
| | Scene-play | 0.1 | 0.001 | 32 | 300 | 0.95 | 0.95 | 100 | 1 | 0.3 | 100 | 3 |
| | Cube-double-play | 0.1 | 0 | 32 | 100 | 0.9 | 0.95 | 300 | 1 | 3 | 100 | 0 |
| | Puzzle-3x3-play | 0.3 | 0.001 | 32 | 1000 | 0.8 | 0.95 | 100 | 0.5 | 0.3 | 100 | 3 |
| | Puzzle-4x4-play | 0.3 | 0.01 | 32 | 1000 | 0.95 | 0.95 | 1000 | 3 | 100 | 100 | 3 |
| OGBench (Pixel-based) | Visual-antmaze-medium-navigate | 0.01 | 0.003 | 32 | 100 | 0.9 | 0.9 | 10 | 0.3 | 0.03 | 10 | 0.1 |
| | Visual-antmaze-teleport-navigate | 0.01 | 0.003 | 32 | 100 | 0.8 | 0.95 | 3 | 0.3 | 0.03 | 10 | 0.3 |
| | Visual-cube-double-play | 0.1 | 0 | 32 | 100 | 0.9 | 0.95 | 100 | 1 | 0.3 | 100 | 0.1 |
| | Visual-puzzle-4x4-play | 0.3 | 0.01 | 32 | 300 | 0.9 | 0.9 | 100 | 1 | 0.3 | 100 | 0.1 |

*Table 5.* **Hyperparameter Configurations for Ablation Studies 1 (Flow Anchoring) and 2 ($\mathcal{T}_n^\pi$).** We use discount factor of $\gamma = 0.995$, and follow the similar hyperparameter choices in Table 4.

| | | NON-DISTRIBUTIONAL | | | DISTRIBUTIONAL | | | | |
| | | FQL | FAQL (FLOW ANCHORING) | | NBRAC (ReBRAC-STYLE BC) | | NFQL (FQL-STYLE BC) | FAN (FLOW ANCHORING) | |
| BENCHMARK | TASK | $\alpha$ | $\alpha_1$ | $\alpha_2$ | $\alpha_1$ | $\alpha_2$ | $\alpha$ | $\alpha_1$ | $\alpha_2$ |
|---|---|---|---|---|---|---|---|---|---|
| OGBENCH (STATE-BASED) | ANTSOCCER-NAVIGATE | 10 | 10 | 0.1 | 10 | 0.1 | 30 | 10 | 0.1 |
| | SCENE-PLAY | 1000 | 100 | 3 | 100 | 3 | 1000 | 100 | 3 |
| | CUBE-DOUBLE-PLAY | 300 | 100 | 0 | 100 | 0 | 300 | 100 | 0 |
| | PUZZLE-3X3-PLAY | 1000 | 100 | 3 | 100 | 3 | 1000 | 100 | 3 |
| | PUZZLE-4X4-PLAY | 1000 | 100 | 3 | 100 | 3 | 1000 | 100 | 3 |

*Table 6.* **Hyperparameter Configurations for Ablation Study 3 (Offline-to-Online).** We use discount factor of $\gamma = 0.995$ for results not present in prior work. "-" indicates that the results are taken from prior work or excluded, and $\rightarrow$ indicates the hyperparameter change for online training.

| | | NON-DISTRIBUTIONAL | | | | DISTRIBUTIONAL | | | | | |
| | | ReBRAC | | IDQL | FQL | IQN | VALUE FLOWS | | | FAN | |
| BENCHMARK | TASK | $\alpha_1$ | $\alpha_2$ | $N$ | $\alpha$ | $\kappa$ | $\lambda$ | $\tau$ | $\alpha$ | $\alpha_1$ | $\alpha_2$ |
|---|---|---|---|---|---|---|---|---|---|---|---|
| OGBENCH (STATE-BASED) | ANTSOCCER-MEDIUM-NAVIGATE | 0.01 | 0.01 | 64 | 30 | 0.9 | 1 | 1 | $0 \rightarrow 30$ | $30 \rightarrow 10$ | $1 \rightarrow 1$ |
| | SCENE-PLAY | 0.1 | 0.01 | 32 | 300 | 0.95 | 1 | 0.3 | $0 \rightarrow -$ | $100 \rightarrow 10$ | $3 \rightarrow 0$ |
| | CUBE-DOUBLE-PLAY | 0.1 | 0 | 32 | 300 | 0.9 | 1 | 3 | $0 \rightarrow -$ | $100 \rightarrow 30$ | $0 \rightarrow 0$ |
| | PUZZLE-3X3-PLAY | 0.3 | 0.01 | 32 | 1000 | 0.8 | 0.5 | 0.3 | $0 \rightarrow 1000$ | $100 \rightarrow 10$ | $3 \rightarrow 0$ |
| | PUZZLE-4X4-PLAY | 0.3 | 0.01 | 32 | 1000 | 0.95 | 3 | 100 | $0 \rightarrow -$ | $100 \rightarrow 100$ | $3 \rightarrow 0$ |

## C.4. Full Results

*Table 7.* **Full Offline Results** on the reward-based OGBench and D4RL tasks stated in Table 1. The results are collected over 8 seeds for state-based and 4 seeds for pixel-based tasks. The numbers are bolded if they are above or equal to 95% of the best performance.

| BENCHMARK | TASK | NON-DISTRIBUTIONAL | | | DISTRIBUTIONAL | | | |
|---|---|---|---|---|---|---|---|---|
| | | REBRAC | IDQL | FQL | IQN | CODAC | VALUE FLOWS | FAN |
| D4RL (ANTMAZE) | ANTMAZE-MEDIUM-PLAY-V2 | **90** | 84 | 78±7 | 38±5 | 82±2 | 16±3 | 82±3 |
| | ANTMAZE-MEDIUM-DIVERSE-V2 | **84** | 85 | 71±13 | 40±4 | 10±2 | 10±5 | 76±3 |
| | ANTMAZE-LARGE-PLAY-V2 | 52 | 64 | **84**±7 | 51±5 | 71±3 | 12±2 | 77±5 |
| | ANTMAZE-LARGE-DIVERSE-V2 | 64 | 68 | **83**±4 | 55±3 | 22±5 | 30±10 | 70±5 |
| D4RL (ADROIT) | PEN-HUMAN-V1 | **103** | 71±12 | 53±6 | 69±3 | 67±0 | 66±4 | 64±11 |
| | PEN-CLONED-V1 | **103** | 80±11 | 74±11 | 80±11 | 76±2 | 73±5 | 90±9 |
| | PEN-EXPERT-V1 | **152** | 139±5 | 142±6 | 118±19 | 136±2 | 117±3 | 138±6 |
| | DOOR-HUMAN-V1 | 0 | 7±2 | 0±0 | 0±0 | 3±1 | 7±2 | **8**±2 |
| | DOOR-CLONED-V1 | 0 | 2±1 | 2±1 | 0±0 | 0±0 | 0±0 | **5**±3 |
| | DOOR-EXPERT-V1 | **106** | 104±2 | 104±1 | 105±0 | 104±0 | 104±1 | 104±1 |
| | HAMMER-HUMAN-V1 | 0 | **3**±1 | 1±1 | 2±1 | **3**±1 | 1±0 | **3**±1 |
| | HAMMER-CLONED-V1 | 5 | 2±1 | **11**±9 | 0±0 | 6±0 | 1±0 | 3±2 |
| | HAMMER-EXPERT-V1 | 134 | 117±9 | 125±3 | 121±7 | 126±1 | 125±5 | 115±5 |
| | RELOCATE-HUMAN-V1 | **0** | **0**±0 | **0**±0 | **0**±0 | **0**±0 | **0**±0 | **0**±1 |
| | RELOCATE-CLONED-V1 | **2** | 0±0 | 0±0 | 0±0 | 0±0 | 0±0 | 0±0 |
| | RELOCATE-EXPERT-V1 | **108** | 104±3 | 107±1 | 103±0 | 103±2 | 102±2 | **106**±1 |
| OGBENCH (STATE-BASED) | ANTSOCCER-NAVIGATE-TASK1 | 0±0 | 61±25 | 77±4 | 30±5 | 24±18 | 56±8 | **89**±4 |
| | ANTSOCCER-NAVIGATE-TASK2 | 0±1 | 75±3 | 88±3 | 14±7 | 63±19 | 39±10 | **91**±7 |
| | ANTSOCCER-NAVIGATE-TASK3 | 0±0 | 14±22 | **61**±6 | 34±12 | 25±8 | 7±3 | 49±8 |
| | ANTSOCCER-NAVIGATE-TASK4 | 0±0 | 16±9 | 39±6 | 27±9 | 32±15 | 21±7 | **49**±8 |
| | ANTSOCCER-NAVIGATE-TASK5 | 0±0 | 0±1 | **36**±9 | 16±5 | 19±4 | 10±7 | 21±14 |
| OGBENCH (STATE-BASED) | SCENE-PLAY-TASK1 | 96±8 | 98±3 | **100**±0 | **100**±0 | 99±0 | 99±0 | **100**±0 |
| | SCENE-PLAY-TASK2 | 50±13 | 0±0 | 76±9 | 1±0 | 85±4 | **97**±1 | **96**±3 |
| | SCENE-PLAY-TASK3 | 78±4 | 54±19 | **98**±1 | 94±2 | 90±3 | 94±2 | 93±4 |
| | SCENE-PLAY-TASK4 | 4±4 | 0±0 | 5±1 | 3±1 | 0±0 | **7**±17 | 0±0 |
| | SCENE-PLAY-TASK5 | 0±0 | 0±0 | 0±0 | 0±0 | 0±0 | 0±0 | 0±0 |
| OGBENCH (STATE-BASED) | CUBE-DOUBLE-PLAY-TASK1 | 47±11 | 35±9 | 61±9 | 70±14 | 80±11 | **97**±1 | 84±5 |
| | CUBE-DOUBLE-PLAY-TASK2 | 22±12 | 9±5 | 36±6 | 24±9 | 63±4 | **76**±7 | 59±10 |
| | CUBE-DOUBLE-PLAY-TASK3 | 4±1 | 8±5 | 22±5 | 25±6 | 66±9 | **73**±4 | 40±17 |
| | CUBE-DOUBLE-PLAY-TASK4 | 1±1 | 1±1 | 5±2 | 10±1 | 13±2 | **30**±5 | 5±5 |
| | CUBE-DOUBLE-PLAY-TASK5 | 4±2 | 17±6 | 19±10 | 81±8 | 82±4 | 69±5 | 43±18 |
| OGBENCH (STATE-BASED) | PUZZLE-3X3-PLAY-TASK1 | 97±4 | 94±3 | 90±4 | 71±3 | 78±8 | 99±0 | **100**±1 |
| | PUZZLE-3X3-PLAY-TASK2 | 1±1 | 1±2 | 16±5 | 2±2 | 5±2 | 98±2 | **100**±0 |
| | PUZZLE-3X3-PLAY-TASK3 | 3±1 | 0±0 | 10±3 | 0±0 | 4±3 | 97±1 | **99**±2 |
| | PUZZLE-3X3-PLAY-TASK4 | 2±1 | 0±0 | 16±5 | 0±0 | 5±5 | 84±24 | **100**±1 |
| | PUZZLE-3X3-PLAY-TASK5 | 5±3 | 0±0 | 16±3 | 0±0 | 6±5 | 58±39 | **99**±2 |
| OGBENCH (STATE-BASED) | PUZZLE-4X4-PLAY-TASK1 | 32±9 | 49±9 | 34±8 | 41±2 | 37±32 | 36±3 | **83**±4 |
| | PUZZLE-4X4-PLAY-TASK2 | 16±4 | 4±4 | 16±5 | 12±4 | 10±10 | **27**±5 | 21±9 |
| | PUZZLE-4X4-PLAY-TASK3 | 20±10 | 50±14 | 18±5 | 45±7 | 33±29 | 30±4 | **81**±13 |
| | PUZZLE-4X4-PLAY-TASK4 | 10±3 | 21±11 | 11±3 | 23±2 | 12±10 | **28**±5 | 12±8 |
| | PUZZLE-4X4-PLAY-TASK5 | 7±3 | 2±2 | 7±3 | **16**±6 | 10±8 | 13±2 | 13±16 |
| OGBENCH (PIXEL-BASED) | VISUAL-ANTMAZE-MEDIUM-TASK1 | 54±15 | 81±3 | 32±3 | 62±7 | **94**±1 | 77±4 | **92**±4 |
| | VISUAL-ANTMAZE-TELEPORT-TASK1 | 2±0 | 7±4 | 2±1 | 2±1 | 3±3 | **10**±4 | 5±3 |
| | VISUAL-CUBE-DOUBLE-PLAY-TASK1 | 6±2 | 8±6 | 23±4 | 4±1 | 3±2 | **35**±2 | **35**±15 |
| | VISUAL-PUZZLE-4X4-PLAY-TASK1 | 26±6 | 8±15 | **33**±6 | 7±4 | 0±0 | 24±5 | 30±16 |

