# OpenReview forum: "Towards Efficient and Expressive Offline RL via Flow-Anchored Noise-conditioned Q-Learning"
_ICML.cc/2026/Conference — ICML 2026 regular_

### Official Review · Reviewer_apDo · 2026-03-12

**Soundness:** 3
**Presentation:** 2
**Significance:** 3
**Originality:** 2
**Overall Recommendation:** 4
**Confidence:** 5

**Summary:**

The paper studies offline RL under an efficiency-performance trade-off. The paper argues that recent expressive approaches based on flow policies and distributional critics improve return quality but are computationally expensive because they require either iterative flow sampling or multiple critic samples. To address this, the paper proposes Flow-Anchored Noise-conditioned Q-Learning, which combines two ideas: 1) flow anchoring, a behavior-regularization mechanism that uses only a single flow iteration instead of full ODE-based behavior sampling; 2) a noise-conditioned critic trained with a single Gaussian noise sample together with an upper-expectile estimator intended to approximate an essential supremum over return noise. The paper also provides convergence and regularization results for the proposed operator and reports experiments on D4RL and OGBench showing competitive or state-of-the-art task performance together with substantially improved training and inference efficiency.

**Compliance With Llm Reviewing Policy:**

Affirmed.

**Final Justification:**

The paper constrains the policy near the data distribution through flow anchoring, providing a new strategy constraint approach for offline reinforcement learning. This design is relevant to the distribution shift issue in offline reinforcement learning. The paper provides empirical evidence for the effectiveness of this component through ablation comparisons with related variants, and also provides theoretical analysis of the convergence and regularization properties of the proposed operator.
The paper is well organized. It also supplements the analysis of hyperparameter selection in the rebuttal and provide additional sensitivity experiments.

**Key Questions For Authors:**

See Weaknesses.

**Limitations:**

yes

**Strengths And Weaknesses:**

Strengths:
1. The paper constrains the policy near the data distribution through flow anchoring, providing a new strategy constraint approach for offline reinforcement learning. This design is relevant to the distribution shift issue in offline reinforcement learning, and the paper provides some empirical evidence for the effectiveness of this component through ablation comparisons with related variants.
2. The paper provides theoretical analysis of the convergence and regularization properties of the proposed operator. This part provides the method with stronger theoretical support compared to purely empirical approaches.
3. The proposed method is evaluated on standard benchmarks such as D4RL and OGBench, and achieves competitive task performance.

Weaknesses:
1. The paper lacks sufficient explanation or analysis for the selection of some key hyperparameters. For example, the expectile coefficient used in the value estimation process.
2. The paper fixes the expectile coefficient $\kappa = 0.9$ for all experiments, while the sensitivity analysis in Appendix D.3 is conducted on only a small number of tasks, leaving the robustness of this choice across different environments unclear.

---

> ### Author Rebuttal · Authors · 2026-03-26
>
> Dear Reviewer apDo,
> \
> We appreciate your time and effort in reviewing our paper.
>
> We believe your main concern is:
>
> - the lack of explanation or analysis regarding our key hyperparameter selections ($\alpha_1,\alpha_2,\kappa$).
>
> Therefore, we elaborate on the explanations and provide further ablation studies.
>
> ---
>
> # Questions
>
> > **Q.1. The paper lacks sufficient explanation or analysis for the selection of some key hyperparameters.**
> >
>
> > **A.1.1 [Explanation]
> \
> (1) $\alpha_1,\alpha_2$: the best values are selected from a parameter sweep, with $\alpha_1$ having $\alpha$ values similar to those used in FQL (Appendix C.2).
> \
> (2) $\kappa$: set to 0.9 (near 1) to realize $\mathcal{T}^\pi_n$ (Theorem 4.2), which is similar to the standard choice in IQL.**
> >
>
> We did a parameter sweep of $\alpha_1,\alpha_2$ within a tuning budget similar to that of prior work.
> \
> In particular, $\alpha_1$ is kept similar to the $\alpha$ values used in FQL, as they are both coefficients for flow-based behavior regularization during policy training.
>
> $\kappa=0.9$ is selected to realize $\mathcal{T}^\pi_n$. This aligns with the standard choice in IQL, which also uses expectile regression.
>
> > **A.1.2 [Analysis] FAN requires $\alpha_1,\alpha_2$ tuning.**
> >
>
> We further analyzed how changing $\alpha_1,\alpha_2$ changes the success rates on four tasks, using 8 seeds and following Tables 3 and 4:
>
> **scene-play-task1**
>
> | $\alpha_1$ \ $\alpha_2$ | 1 | 3 | 10 |
> | --- | --- | --- | --- |
> | **30** | **100**$\pm1$ | $91\pm2$ | $0\pm0$ |
> | **100** | **100**$\pm0$ | **100**$\pm0$ | **99**$\pm1$ |
> | **300** | **98**$\pm1$ | **100**$\pm0$ | **100**$\pm0$ |
>
> **cube-double-play-task1**
>
> | $\alpha_1$ \ $\alpha_2$ | 1 | 3 | 10 |
> | --- | --- | --- | --- |
> | **30** | $38\pm6$ | $36\pm2$ | $0\pm0$ |
> | **100** | $80\pm10$ | **89**$\pm6$ | $30\pm3$ |
> | **300** | $59\pm7$ | $68\pm3$ | $78\pm4$ |
>
> **puzzle-3x3-play-task1**
>
> | $\alpha_1$ \ $\alpha_2$ | 1 | 3 | 10 |
> | --- | --- | --- | --- |
> | **30** | **100**$\pm0$ | **98**$\pm1$ | $44\pm19$ |
> | **100** | **100**$\pm0$ | **100**$\pm1$ | **100**$\pm0$ |
> | **300** | **99**$\pm1$ | **100**$\pm1$ | **100**$\pm0$ |
>
> **puzzle-4x4-play-task1**
>
> | $\alpha_1$ \ $\alpha_2$ | 1 | 3 | 10 |
> | --- | --- | --- | --- |
> | **30** | $34\pm5$ | $56\pm6$ | $2\pm1$ |
> | **100** | $76\pm8$ | **83**$\pm4$ | $66\pm10$ |
> | **300** | $67\pm9$ | $74\pm3$ | $76\pm3$ |
>
> When changing $\alpha_1$, the mean changes in the average success rates are 19, 36, 10, and 27. When changing $\alpha_2$, they are 17, 21, 10, and 18.
> \
> Therefore, we conclude that FAN is sensitive to these coefficients, and $\alpha_1$ has a stronger effect.
> \
> However, we would like to highlight that such sensitivity is typical for most offline RL methods [1].
>
> During our study, $\alpha_1$ is selected from FQL, and $\alpha_2$ values are maintained between 0 and 10 for all task environments.
> \
> Hence, our search spaces for $\alpha_1,\alpha_2$ were not significantly extensive compared to prior work.
>
> Please refer to the four additional sensitivity analyses provided in our rebuttal to Reviewer tGLN.
>
> ---
>
> > **Q.2. The sensitivity analysis in Appendix D.3 is conducted on only a small number of tasks.**
> >
>
> > **A.2. $\kappa=0.9$ is good across a broad range of tasks.**
> >
>
> For a clearer view, we trained FAN on 7 additional tasks with $\kappa\in[0.5,0.7,0.9,0.99,1.0]$:
>
> | $\kappa$ | d4rl antmaze-medium-play | d4rl antmaze-medium-diverse | d4rl antmaze-large-play | d4rl antmaze-large-diverse | scene-play-task1 | cube-double-play-task1 | puzzle-3x3-play-task1 |
> | --- | --- | --- | --- | --- | --- | --- | --- |
> | 0.5 | $74\pm3$ | $65\pm1$ | $23\pm4$ | $21\pm7$ | **100**$\pm0$ | $83\pm2$ | $99\pm1$ |
> | 0.7 | $77\pm6$ | $73\pm5$ | **78**$\pm1$ | **74**$\pm2$ | **100**$\pm0$ | $22\pm8$ | **100**$\pm0$ |
> | 0.9 | **82**$\pm3$ | **76**$\pm3$ | $77\pm5$ | $70\pm5$ | **100**$\pm0$ | **84**$\pm5$ | **100**$\pm1$ |
> | 0.99 | $7\pm2$ | $28\pm3$ | $4\pm2$ | $5\pm3$ | $98\pm1$ | $19\pm6$ | $94\pm3$ |
> | 1.0 | $0\pm0$ | $0\pm0$ | $0\pm0$ | $0\pm0$ | $14\pm6$ | $5\pm1$ | $2\pm1$ |
>
> The table shows that **$\kappa=0.9$ remains the best choice**, as it achieved the best results on 6 out of 9 tasks, including the two tasks in our original submission.
> \
> $\kappa=0.7$ is also a competitive choice, achieving the best results on 5 out of 9 tasks.
> \
> We believe it may be possible to find a better $\kappa$ value between 0.7 and 0.99, and this tuning range aligns well with IQL.
> \
> However, in our case, we believe $\kappa=0.9$ should be treated similarly to the learning rate (3e-4) and the target value update rate (0.005), since they are all fixed across all tasks.
>
> To summarize, we would like to highlight that **FAN is highly effective even though we tune only two coefficients: $\alpha_1$ and $\alpha_2$.**
>
> ---
>
> We hope our rebuttal has addressed your concerns, and we appreciate your thorough review.
>
> ### References
>
> [1] Tarasov et al., CORL: Research-oriented deep offline reinforcement learning library, NIPS23

---

> > ### Author Rebuttal · Reviewer_apDo · 2026-04-02
> >
> > The authors supplement the analysis of hyperparameter selection in the rebuttal and provide additional sensitivity experiments, which meet my expectations. Based on this, I will revise the Overall Recommendation of this paper to Weak Accept.

---

> > > ### Author Response · Authors · 2026-04-02
> > >
> > > Dear Reviewer apDo,
> > > \
> > > We deeply appreciate your positive view of our work,
> > > \
> > > and thank you for acknowledging our additional sensitivity experiments and for raising your score.
> > >
> > > Best regards,
> > > \
> > > The Authors

---

### Official Review · Reviewer_tGLN · 2026-03-12

**Soundness:** 3
**Presentation:** 2
**Significance:** 3
**Originality:** 3
**Overall Recommendation:** 5
**Confidence:** 4

**Summary:**

The manuscript proposes Flow-Anchored Noise-conditioned Q-Learning (FAN) with the goal of mitigating current problems in flow-based offline policies, such as the overestimation of Q-values for out-of-distribution actions, while simultaneously improving the inference latency originating from the policies’ iterative sampling nature. To achieve this, first, the authors approach value estimation from a distributional RL perspective by conditioning the critic network on a single noise sample, coupled with an upper expectile regressor. Through this, FAN implicitly models the distribution of returns while capturing the maximum possible returns. Additionally, FAN utilizes 'Flow Anchoring' in both the actor and critic models for behavior regularization. This acts as a single-step distillation of the flow model's vector field at a uniformly sampled timestep, avoiding the need to simulate the full ODE. FAN is evaluated on the OGBench and D4RL benchmarks, where it performs better than the majority of the baselines while demonstrating inference times that are competitive with non-distributional variants.

**Compliance With Llm Reviewing Policy:**

Affirmed.

**Final Justification:**

The rebuttal addressed my concerns. I believe this work is a solid contribution, and thus, I will maintain my recommendation to accept.

**Key Questions For Authors:**

1. Could the authors provide a sensitivity experiment for $\alpha_1$ and $\alpha_2$? How much does performance degrade if a single, fixed set of hyperparameters is used across all state-based OGBench or D4RL tasks?

**Limitations:**

The authors could have discussed limitations such as the hyperparameter sensitivity mentioned before, as well as the training stability involved in optimizing multiple coupled models.

**Strengths And Weaknesses:**

### Strengths

1. The core problem of the manuscript is important, especially,  nowadays with the increasing adoption of generative policies in robotics. Thus, the Flow Anchoring idea is an interesting contribution to reducing the training time.
2. Unlike prior methods (such as FQL) that require solving the ODE during training, FAN regularizes the policy directly against the flow's velocity field. This is a nice way to bypass ODE integration entirely during training.
3. The approach is theoretically grounded, which effectively supports the design choices of FAN. Additionally, it provides promising results given the extensive evaluation, especially in OGBench, which includes some hard to solve tasks.

### Weaknesses

1. In my opinion, an important flaw of the paper is the writing. The manuscript does not have a coherent flow, and lacks a lot of important information, vital for the reader’s understanding. The manuscript dives directly in the theoretical formulation of the approach without first providing any intuition about the choices that have been made. To be more specific, the information in Appendix A.3, is a clear statement of the motivation and intuition (e.g., how the critic acts as an implicit distribution representation). This should have been in the main text, since it provides a clear view of the proposed solution, before diving into the details.
2. Table 4 shows that the a_1 and a_2 parameters have a lot of variability. This raises the concern of the stability of the approach and how sensitive it is regarding the hyperparameters. A sensitivity analysis of FAN would have been much appreciated.
3. While Flow Anchoring is effective, the authors do not adequately position it within the broader flow/diffusion policy distillation literature (which frequently utilizes velocity-matching techniques). This connection would strengthen the paper's specific novelty claims.

---

> ### Author Rebuttal · Authors · 2026-03-29
>
> Dear Reviewer tGLN,
> \
> We appreciate your time and effort, and your positive view of our work.
>
> We believe your main concerns are:
> \
> (1) the flow of the writing; and
> \
> (2) the sensitivity analysis on $\alpha_1,\alpha_2$.
>
> ---
>
> # Questions
>
> > **Q.1. The manuscript dives directly in the theoretical formulation without first providing any intuition about the choices that have been made.**
> >
>
> > **A.1. We will move the “Motivation” paragraphs in Sections 4.1/4.2 to before the “Notations and Function Definitions” paragraph. This will provide an intuitive explanation of the key ideas in the paper  first.**
> >
>
> We have realized that it is not optimal to place the “Notations and Function Definitions” immediately after the “Main Focus."
> \
> Therefore, we will reorganize the “Motivation” paragraphs in Sections 4.1/4.2 so they appear between the “Main Focus” and the “Notations and Function Definitions” paragraphs.
>
> Moreover, we believe the following sentences might be helpful:
>
> - Our intuition is that all critic samples with this noise-conditioned Q function have a similar meaning, in that each is a possible return outcome sampled randomly. Consequently, using a single critic sample may be sufficient.
> - Our intuition is that accurate sampling of the dataset actions is actually not a strict requirement for optimal policy learning. These actions are merely used for regularizing the optimal policy.
>
> ---
>
> > **Q.2. The authors do not position FAN within the broader flow/diffusion policy distillation literature.**
> >
>
> > **A.2. If you find it appropriate, we propose adding sentences addressing the policy distillation literature in Section 2 (Related Work).**
> >
>
> Due to page limits, we initially chose to provide a high-level overview of flow policies and distributional critics. However, our method is indeed highly related to policy distillation.
>
> Therefore, as the second-to-last sentence of the “Diffusion and Flow Policies in Offline RL” paragraph in Section 2, we would like to suggest adding the following:
>
> - In particular, for behavior regularization, diffusion and flow policies normally distill offline behaviors into learning the optimal policies [1], also including approaches using few-step generative modeling [2][3]. However, their inference efficiency comes at the cost of additional computations during training.
>
> ---
>
> > **Q.3. Could the authors provide a sensitivity experiment for $\alpha_1$ and $\alpha_2$? How much does performance degrade if a single, fixed set of hyperparameters is used across diverse tasks?**
> >
>
> > **A.3. As is typical for most offline RL methods [4], FAN is sensitive to $\alpha_1$ and $\alpha_2$, with $\alpha_1$ having a greater impact. However, strong performance can be maintained even with a fixed set of hyperparameters across different task environments.**
> >
>
> We provide a sensitivity analysis on four default task environments in OGBench, using 8 seeds and following Table 3:
>
> **scene-play**
>
> | $\alpha_1$ \ $\alpha_2$ | 1 | 3 | 10 |
> | --- | --- | --- | --- |
> | **30** | $86\pm10$ | $75\pm12$ | $0\pm0$ |
> | **100** | **99**$\pm2$ | **96**$\pm3$ | **97**$\pm2$ |
> | **300** | $74\pm10$ | **95**$\pm2$ | **100**$\pm0$ |
>
> **cube-double-play**
>
> | $\alpha_1$ \ $\alpha_2$ | 1 | 3 | 10 |
> | --- | --- | --- | --- |
> | **30** | $26\pm5$ | $11\pm4$ | $0\pm0$ |
> | **100** | $52\pm10$ | **59**$\pm10$ | $30\pm2$ |
> | **300** | $24\pm6$ | $33\pm5$ | $31\pm10$ |
>
> **puzzle-3x3-play**
>
> | $\alpha_1$ \ $\alpha_2$ | 1 | 3 | 10 |
> | --- | --- | --- | --- |
> | **30** | $90\pm5$ | $82\pm2$ | $0\pm0$ |
> | **100** | **95**$\pm2$ | **100**$\pm1$ | **99**$\pm2$ |
> | **300** | $36\pm2$ | $74\pm1$ | **100**$\pm0$ |
>
> **puzzle-4x4-play**
>
> | $\alpha_1$ \ $\alpha_2$ | 1 | 3 | 10 |
> | --- | --- | --- | --- |
> | **30** | $8\pm3$ | $2\pm1$ | $0\pm0$ |
> | **100** | $22\pm10$ | $12\pm8$ | $9\pm2$ |
> | **300** | **28**$\pm4$ | $15\pm2$ | $10\pm3$ |
>
> When changing $\alpha_1$, the mean changes in the average success rates are 27, 27, 35, and 7, and when changing $\alpha_2$, they are 19, 12, 27, and 7.
> \
> Hence, FAN is sensitive to both, but is more sensitive to $\alpha_1$.
>
> However, strong performance is maintained across the four tasks when using a fixed setting of either $(\alpha_1,\alpha_2)=(100,1)$ or $(100,3)$. Specifically, $(\alpha_1,\alpha_2)=(100,1)$ yields average success rates of 99, 52, 95, and 22. Also, at $(100,3)$, the success rates remain highly competitive at 96, 59, 100, and 12. This demonstrates that a single set of hyperparameters for FAN can achieve near peak performance across diverse environments.
>
> Please refer to the four additional sensitivity analyses provided in our rebuttal to Reviewer apDo.
>
> ---
>
> We hope we have addressed your concerns, and we thank you again for your insightful feedback.
>
> ### References
>
> [1] Zhang et al., ReFORM, ICLR26.
>
> [2] Zhan and Tao et al., MVP, ICLR26.
>
> [3] Koirala et al., SSCP, ICLR26.
>
> [4] Park et al., Is value learning really the main bottleneck in offline rl?, NIPS24

---

> > ### Author Rebuttal · Reviewer_tGLN · 2026-04-03
> >
> > I thank the authors for their thorough response. The additional experiments regarding hyperparameter sensitivity effectively address my main concerns. Furthermore, the restructured motivation section will significantly enhance the manuscript's narrative flow and clarity. Overall, this is a solid paper, and I will maintain my recommendation to accept.

---

> > > ### Author Response · Authors · 2026-04-03
> > >
> > > Dear Reviewer tGLN,
> > > \
> > > We deeply appreciate your positive feedback and strong support for our work.
> > > \
> > > Your insights have been incredibly helpful, and we will revise the manuscript to reflect our discussion in the rebuttal.
> > >
> > > Thank you again for your time and effort.
> > > \
> > > Best regards,
> > > \
> > > The Authors

---

### Official Review · Reviewer_v6hk · 2026-03-12

**Soundness:** 3
**Presentation:** 3
**Significance:** 4
**Originality:** 4
**Overall Recommendation:** 5
**Confidence:** 4

**Summary:**

This paper proposes a flow matching-based offline RL algorithm that seeks to improve the computational efficiency of (1) flow-based behavioral constraints and (2) distributional critics.

The actor constraint consists of training a behavior cloning flow policy $\pi_\theta$ using standard flow matching for a behavioral constraint on a separate one-step policy $\pi_\omega$, similar to prior work (FQL). However, rather than generating the full action from $\pi_\theta$ from sampled noise $\epsilon$, they instead construct an interpolant from $\epsilon$ and $a_\omega$ from the one-step policy, and query $v_\theta$ for a single velocity vector instead of a full trajectory. Then, they regularize $\pi_\omega$ towards the output of $v_\theta(s, t, a_{t, \omega})$.

The distributional critic is trained in a similar “flow”-like manner, where the network itself outputs a deterministic scalar, but takes in random noise $\epsilon$ as an input to produce a distribution over returns. Similar to standard expectile regression, we maintain a critic and a “baseline” used to construct the Bellman target. This baseline $Z_\psi(s, a_\omega)$ is trained with expectile regression towards the maximum values of $Q(s,a_\omega, \epsilon)$, which effectively performs expectile regression towards the upper quantiles of the distribution parameterized by $Q(s,a_\omega, \epsilon)$.

They benchmark this method on offline RL and offline-to-online RL, showing state-of-the-art results for locomotion and manipulation in both settings

**Compliance With Llm Reviewing Policy:**

Affirmed.

**Final Justification:**

My primary concerns (some ablations and choice of tasks) were addressed. I think the noise-conditioned critic will be of significant interest to the community, and will inspire additional work on the effectiveness of this distributional approach in O2O. Therefore, I update my score to a 5 (accept).

**Key Questions For Authors:**

1. FQL does not use the BC penalty in the critic, and I would be interested to see an ablation that removes the critic penalty and isolates the effectiveness solely of Flow Anchoring, e.g., substituting it directly into FQL without other changes.
2. I wonder if the Anchoring objective provides a somewhat looser constraint than regularizing towards full action generations? Intuitively, enforcing the BC penalty to only one intermediate sample of the velocity may push the action in a different direction than any action in the $\pi_\mathrm{BC}$ distribution, especially if the flow field is highly curved.
3. I am also curious about the authors’ theories on the significant performance improvements in O2O — is it due to better exploration (perhaps due to a looser BC constraint as above), improved stability of the distributional critic, or something else?
4. The choice of tasks seems somewhat arbitrary — can the authors explain why they specifically chose `antsoccer` and omitted all other navigation tasks, and why they only evaluated `cube-double` specifically out of all the cube 1-4 tasks? Also, is there are reason they prefer play datasets over the noisy ones for manipulation? What is performance like for noisy datasets?
5. Why is the inference time of the (distributional) value function as shown in Figure 3 useful? To my understanding, FAN is not doing rejection sampling, so wouldn’t the policy inference time be the only quantity that matters?

**Limitations:**

Yes

**Strengths And Weaknesses:**

**Strengths**

The idea of parameterizing a distributional critic using noise conditioning $\epsilon$ is elegant and (to my limited knowledge) novel. The empirical results are strong, especially for O2O transfer. Overall, I think this is a very nice paper and the noise-conditioned critic is especially interesting.

**Weaknesses**

My main concerns are with respect to the evaluation and some ablations. The choice of tasks seems fairly ad-hoc, and I would like to see more comprehensive coverage across the manipulation suites. I’m particularly curious about how this method would perform with the noisy manipulation datasets, which are generally stronger for reparameterized gradient-based methods like FQL/FAC. Additionally, according to the ablations in Figure 4, it appears that the Flow Anchoring component can be significantly worse than a multi-step constraint (FAQL), especially in ``cube-double``. I would like to see more controlled ablations (see Q1).

---

> ### Author Rebuttal · Authors · 2026-03-29
>
> Dear Reviewer v6hk,
> \
> We appreciate your time and effort, particularly your note that our noise-conditioned critic idea is elegant and novel.
>
> We believe your main concerns are:
> \
> (1) the lack of evaluation on other datasets, including the noisy manipulation tasks in OGBench; and
> \
> (2) the lack of ablation on Flow Anchoring with $\alpha_2=0$ (equivalent to directly substituting FQL).
>
> Therefore, we provide additional experiments using 8 seeds, following Tables 3 to 6.
>
> ---
>
> # Questions
>
> > **Q.1. What is performance like for OGBench noisy manipulation datasets?**
> >
>
> > **A.1. FAN performs the best in all four additional noisy task environments.**
> >
>
> We compared FQL, Value Flows, and FAN on four noisy manipulation tasks, with $\alpha_1=100$ and $\alpha_2=3$ for FAN.
>
> |  | scene-noisy-task1 | cube-double-noisy-task1 | puzzle-3x3-noisy-task1 | puzzle-4x4-noisy-task1 |
> | --- | --- | --- | --- | --- |
> | FQL | $98\pm1$ | $11\pm2$ | $23\pm3$ | $13\pm5$ |
> | Value Flows | **100**$\pm0$ | $95\pm2$ | **100**$\pm0$ | $13\pm1$ |
> | FAN | **100**$\pm0$ | **96**$\pm3$ | **100**$\pm0$ | **60**$\pm4$ |
>
> The table shows that FAN performs the best, with particularly outstanding results in puzzle-4x4, demonstrating a more than **4x improvement**.
>
> Moreover, in offline-to-online settings, **FAN achieves a near 100% success rate**:
>
> |  | cube-double-noisy-task1 | puzzle-4x4-noisy-task1 |
> | --- | --- | --- |
> | $(\alpha_1,\alpha_2)$ | (100,3)→(30,0) | (100,3)→(30,0) |
> | FAN | **97**$\pm4$ | **100**$\pm0$ |
>
> Thank you for introducing the noisy manipulation datasets, as they highlight FAN’s effectiveness.
>
> Regarding our task choices, we mainly followed the tasks used in the work of Value Flows for fair comparison, which is the most recently proposed distributional method for offline RL. We leave evaluations on other datasets for future work.
>
> ---
>
> > **Q.2. In Figure 4, it appears that the Flow Anchoring component can be significantly worse than a multi-step constraint (FAQL), especially in cube-double.**
> >
>
> > **A.2. In Figure 4 for cube-double, the curve compares FAN and FAQL, and they both use Flow Anchoring. The worse performance of FAQL comes from using the standard Q function rather than the proposed noise-conditioned critic.**
> >
>
> We believe there might be a misunderstanding regarding this point. We would like to ask if we have misinterpreted your concern.
>
> ---
>
> > **Q.3. FQL does not use the BC penalty in the critic, and I would be interested to see an ablation that directly substitutes FQL with Flow Anchoring.**
> >
>
> > **A.3. When using $Q(s,a)$, Flow Anchoring with $\alpha_2=0$ *does not improve* final performance over FQL.**
> >
>
> We compare FQL and FAQL following Table 5:
>
> | methods | antsoccer-arena-navigate-task1 | scene-play-task1 | cube-double-play-task1 | puzzle-3x3-play-task1 | puzzle-4x4-play-task1 |
> | --- | --- | --- | --- | --- | --- |
> | FQL | **77**$\pm4$ | **100**$\pm0$ | **61**$\pm9$ | $90\pm4$ | 34$\pm8$ |
> | FAQL $(\alpha_2=0)$ | $53\pm4$ | $99\pm1$ | $53\pm2$ | **100**$\pm0$ | **62**$\pm3$ |
> | FAQL $(\alpha_2\neq0)$ | $63\pm6$ | **100**$\pm0$ | $56\pm6$ | **100**$\pm0$ | **66**$\pm5$ |
>
> Using the standard Q function, it is hard to say that Flow Anchoring with $\alpha_2=0$ improves performance.
> \
> This aligns with your insight that the anchoring objective provides a looser constraint, potentially degrading performance if the flow field is highly curved.
>
> ---
>
> > **Q.4. I am also curious about the authors’ theories on the significant performance improvements in O2O.**
> >
>
> > **A.4. We believe it is due to the noise-conditioned critic.**
> >
>
> Unlike Flow Anchoring, the noise-conditioned critic is not strictly limited to offline or off-policy RL settings.
> \
> We also observed the following in the cube-double-play default task in the O2O setting:
>
> |  | cube-double-play (default) |
> | --- | --- |
> | $(\alpha_1,\alpha_2)$ | (100,0)→(30,0) |
> | FQL | $92\pm3$ |
> | FAQL | $83\pm5$ |
> | FAN | **98**$\pm2$ |
>
> Therefore, we believe our proposed critic has consistently improved O2O results.
>
> ---
>
> > **Q.5. Why is the inference time of the (distributional) value function as shown in Figure 3 useful?**
> >
>
> > **A.5. To mitigate confusion, we suggest changing the term in Figure 3 from
> “per function call” to “per training step (including both the value and the policy) and inference step (single-step action sampling)”.**
> >
>
> The inference time in Figure 3 *does not* directly indicate the value function call time. It is the runtime for a “single-step action sampling”.
> \
> For example, during action sampling, Value Flows uses rejection sampling and calls the value function, but FAN only calls the one-step policy and does not call the value function at all.
> \
> Therefore, in Figure 3, the inference time for FAN is a one-step policy call runtime without the noise-conditioned critic.
>
> ---
>
> We appreciate your positive comments on our work once again and hope we have addressed your concerns.

---

> > ### Author Rebuttal · Reviewer_v6hk · 2026-04-04
> >
> > Thanks for your response, I appreciate the additional experiments on the noisy environments and clarifications on Figure 3, which are useful. I also apologize for my misinterpretation in Figure 4, and my concerns there have been resolved. I think the noise-conditioned critic will be of significant interest to the community and will update my score to a 5 (accept).

---

> > > ### Author Response · Authors · 2026-04-04
> > >
> > > Dear Reviewer v6hk,
> > > \
> > > We are grateful for your positive feedback and strong support for our work.
> > > \
> > > Your points in the discussion have been greatly helpful.
> > >
> > > We appreciate your time and effort again.
> > > \
> > > Best regards,
> > > \
> > > The Authors

---

### Official Review · Reviewer_pbXE · 2026-03-13

**Soundness:** 3
**Presentation:** 3
**Significance:** 3
**Originality:** 3
**Overall Recommendation:** 4
**Confidence:** 3

**Summary:**

This paper proposed a noise-conditioned Q-learning framework to avoid iterative sampling process of a flow policy, which improves the computational efficiency while maintain the expressiveness.

**Compliance With Llm Reviewing Policy:**

Affirmed.

**Key Questions For Authors:**

- The motivation for introducing the noise-conditioned critic $Q(s,a,\varepsilon)$ is reasonable. However, it is still a little bit unclear about its conceptual relation to the standard Q-function $Q(s,a)$. That is, it would be helpful if the paper can clarify whether $Q(s,a,\epsilon)$ can be transformed (or degenerate) to standard Q-function in some cases, or they are totally two different objects.

- The efficiency improvement seems mainly achieved by one-step policy related design (one-step policy + flow anchoring), this part with a *standard Q-learning* (FAQL in ablation) seems gain similar results to FAN, is the more complex noise-conditional Q-learning necessary?

**Limitations:**

- The framework utilize the one-step flow to achieve the goal of this work, however, as mentioned in [1], direct one-step flow seems not a perfect alternative of a entire flow policy. But it seems that if the method in this work utilize the iterative policy, the advantage will loss.

[1]. Pan C, Anantharaman G, Huang N C, et al. Much Ado About Noising: Dispelling the Myths of Generative Robotic Control[J]. arXiv preprint arXiv:2512.01809, 2025.

**Strengths And Weaknesses:**

Strengths:
- The motivation is clear and the design is reasonable. The decomposition into one-step policy, noise-conditioned Q-learning and flow anchoring is easy to follow.
- The core contribution of efficiency while maintaining performance is demonstrated by the experiments.
- The idea is good, utilize a noise-conditioned design to avoid iterative flow policy sampling process.

Weaknesses:
- While experiments can demonstrate the key contribution of this work, the improvement is not so significant.
- The key efficiency design of the method is replacing the action used in actor/critic loss from ODE results into one-step policy results. However, the dominated efficiency issue in training a flow/diffusion policy comes from the backpropagation through whole generating process rather than the no-grad sampling process. In this sense, the efficiency improvement seems not so significant.

---

> ### Author Rebuttal · Authors · 2026-03-30
>
> Dear Reviewer pbXE,
> \
> We appreciate the time and effort you have taken to review our work.
>
> We believe your main concerns are:
> \
> (1) small improvements relative to baselines; and
> \
> (2) unclear effects of the proposed noise-conditioned critic.
>
> To address these, we provide quantitative evidence of FAN’s significance and conduct additional ablations using 8 seeds (following Tables 3 to 5).
>
> ---
>
> # Questions
>
> > **Q.1. the improvement is not significant**
> >
>
> > **A.1.1. [Success Rates] *>3x* better task case counts compared to FQL (Table 7).**
> >
>
> FAN achieves better performance than FQL on 34 out of 45 tasks. We believe this improvement is significant, particularly considering FAN's exceptional efficiency, similar to that of FQL.
> \
> Moreover, we recently demonstrated additional improvements on the OGBench noisy manipulation datasets. Please refer to A.1 in our rebuttal to Reviewer v6hk.
>
> > **A.1.2. [Efficiency] *>5x* faster training, *>47x* faster inference compared to Value Flows (Figure 3).**
> >
>
> Value Flows is the best-performing baseline.
> \
> The efficiency improvement of FAN over Value Flows is outstanding, considering FAN also outperforms in success rates.
>
> ---
>
> > **Q.2. The efficiency improvement seems insignificant since it only solves the no-grad sampling process.**
> >
>
> > **A.2. Flow Anchoring achieves a 30\% decrease in training runtime.**
> >
>
> FQL tackled the inefficiency of backpropagation through the flow action generation process by bringing in the key idea of no-grad sampling.
> \
> FAN further improves it with Flow Anchoring.
>
> To quantify the significance of Flow Anchoring, we compared FQL and FAQL on the single-step training runtime in cube-double-play.
> \
> As a result, **FAQL takes 0.64ms, which is about 30\% less than FQL (0.91ms).**
> \
> This improvement is significant, considering the volume of offline RL experiments typically conducted with more than 8 seeds.
>
> ---
>
> > **Q.3. Relation between the noise-conditioned critic and the standard Q-function**
> >
>
> > **A.3. $\mathbb{E}_{\epsilon}[Q(s,a,\epsilon)]=Q(s,a)$**
> >
>
> We will explain better the relation between $Q(s,a,\epsilon)$ and $Q(s,a)$ by adding the following sentence right after Eq.(9):
>
> - Note that $\mathbb{E}_{\epsilon\sim\mathcal{N}(0,I_d)}[Q(s,a,\epsilon)]$ becomes the standard action value function modeling the expected future return.
>
> ---
>
> > **Q.4. FAQL seems to gain similar results to FAN. Is noise-conditioned Q-learning necessary?**
> >
>
> > **A.4. Noise-conditioned Q-Learning yields better results on 4 out of 5 task environments (averaged over 25 tasks).**
> >
>
> We compared FAQL and FAN on five OGBench task environments (each averaged over tasks 1 to 5):
>
> |  | antsoccer-arena-navigate | scene-play | cube-double-play | puzzle-3x3-play | puzzle-4x4-play |
> | --- | --- | --- | --- | --- | --- |
> | $(\alpha_1,\alpha_2)$ | (10,0.1) | (100,3) | (100,0) | (100,3) | (100,3) |
> | FAQL | $47\pm5$ | **74**$\pm6$ | $22\pm4$ | $99\pm1$ | $15\pm3$ |
> | FAN | **60**$\pm8$ | $58\pm1$ | **46**$\pm11$ | **100**$\pm1$ | **42**$\pm10$ |
>
> The noise-conditioned Q-Learning clearly helps in improving the success rates, achieving better results on 4 out of 5 environments.
>
> ---
>
> > **Q.5. A one-step flow seems not a perfect alternative to an entire flow policy [1].**
> >
>
> > **A.5.1. Settings differ between [1] and our work.**
> >
>
> We thank the reviewer for bringing this work to our attention. It analyzes flow/diffusion policies and is clearly related to FAN, so we will add a reference to this work.
> \
> However, it is difficult to directly apply [1]’s conclusions to our work for the following reasons:
> \
> First, [1] analyzes Behavior Cloning (BC), which is distinct from offline RL.
> \
> Second, they use advanced architectures (e.g., VLA), which are much more expressive than our 5-layer MLPs. These expressive models without flows could have sufficiently covered the effect of flow matching in their BC setup.
>
> > **A.5.2. In theory, one-step flows can model all multi-step flow policies.**
> >
>
> One-step flows can be viewed as modeling the push-forward functions of the multi-step flow policies [2].
> \
> Therefore, with a sufficient training algorithm, one-step flows can represent all possible functions modeled by multi-step flow policies.
>
> > **A.5.3. Inference of one-step flows is at least $N$ times faster.**
> >
>
> Since one-step flow policies only iterate the policy once, they are $N$ times faster than flow policies requiring $N$ iterations per action.
> \
> Moreover, the efficiency increases almost quadratically compared to rejection sampling-based methods, which require multiple action candidates plus value function calculations.
>
> ---
>
> We appreciate your insightful comments on our work and hope we have addressed your concerns.
>
> ### Reference
>
> [1] Pan et al., Much Ado About Noising: Dispelling the Myths of Generative Robotic Control, ICLR26
>
> [2] Deng et al., Generative Modeling via Drifting, arXiv preprint

---

### Decision · Program_Chairs · 2026-04-30

**Decision:**

Accept (regular)

**Comment:**

All the reviewers are positive about the paper's soundness and significance: the motivation is clear, idea is clear and novel, and experimental results are strong. Rebuttal response has addressed initial concerns about writing and ablation studies. Therefore, I recommend the acceptance of this paper.